# Differentiated glioma cell-derived fibromodulin activates integrin-dependent Notch signaling in endothelial cells to promote tumor angiogenesis and growth

Shreoshi Sengupta[1], Mainak Mondal[1], Kaval Reddy Prasasvi[1], Arani Mukherjee[1], Prerna Magod[2], Serge Urbach[3], Dinorah Friedmann-Morvinski[2,4]*, Philippe Marin[3]*, Kumaravel Somasundaram[1]*

[1]Department of Microbiology and Cell Biology, Indian Institute of Science Bangalore, Bangalore, India; [2]School of Neurobiology, Biochemistry and Biophysics, The George S. Wise Faculty of Life Sciences, Tel Aviv, Israel; [3]Institut de Génomique Fonctionnelle, Université de Montpellier, CNRS, INSERM, Montpellier, France; [4]Sagol School of Neuroscience, Tel Aviv University, Tel Aviv, Israel

*For correspondence:
dino@tauex.tau.ac.il (DF-M);
philippe.marin@igf.cnrs.fr (PM);
skumar1@iisc.ac.in (KS)

Competing interest: The authors declare that no competing interests exist.

**Abstract** Cancer stem cells (CSCs) alone can initiate and maintain tumors, but the function of non-cancer stem cells (non-CSCs) that form the tumor bulk remains poorly understood. Proteomic analysis showed a higher abundance of the extracellular matrix small leucine-rich proteoglycan fibro-modulin (FMOD) in the conditioned medium of differentiated glioma cells (DGCs), the equivalent of glioma non-CSCs, compared to that of glioma stem-like cells (GSCs). DGCs silenced for FMOD fail to cooperate with co-implanted GSCs to promote tumor growth. FMOD downregulation neither affects GSC growth and differentiation nor DGC growth and reprogramming in vitro. DGC-secreted FMOD promotes angiogenesis by activating integrin-dependent Notch signaling in endothelial cells. Furthermore, conditional silencing of *FMOD* in newly generated DGCs in vivo inhibits the growth of GSC-initiated tumors due to poorly developed vasculature and increases mouse survival. Collectively, these findings demonstrate that DGC-secreted FMOD promotes glioma tumor angiogenesis and growth through paracrine signaling in endothelial cells and identifies a DGC-produced protein as a potential therapeutic target in glioma.

## Editor's evaluation

The authors shed light on the role that differentiated glioma cells exerts in promoting cancer progression, revealing that the secreted fibromodulin by differentiated glioma cells is crucial in mediating angiogenesis in glioma via integrin-dependent Notch signaling. The results are important for gaining insight into the less concerned differentiated glioma cells in promoting cancer and would potentially enrich the treatment strategy for glioma.

## Introduction

Tumors and their microenvironment form an ecosystem with many cell types that support tumor growth. The key constituents of this ecosystem include cancer stem cells (CSCs), non-cancer stem cells (non-CSCs) representing the differentiated cancer cells, and various other cell types present in

the tumor stroma (*Prager et al., 2019*). It is well established that the tumor-initiating capacity lies solely with CSCs, thereby making them the crucial architects of tumor–stroma interactions that favor tumor growth and progression (*Rheinbay et al., 2013*). CSCs have a dichotomous division pattern as they are capable of self-renewal and give rise to differentiated cells that form the bulk of the tumor (*Olmeda and Ben Amar, 2019*). The indispensable role of CSCs, which usually constitute only a minority population within tumors, is well documented in many solid tumors (*Galli et al., 2004*; *Ignatova et al., 2002*; *Singh et al., 2004*; *Yang et al., 2020a*).

The tumor microenvironment is a vital driver of plasticity and heterogeneity in cancer (*Carnero and Lleonart, 2016*; *Heddleston et al., 2010*). The presence of hypoxic and necrotic regions is the hallmark of very aggressive tumors like glioblastoma (GBM), which have a highly vascular niche that supplies nutrients to cancer cells and makes a conducive environment for the tumor cells to thrive (*Hambardzumyan and Bergers, 2015*; *Huang et al., 2016*). Paracrine signaling mediated by proteins secreted from tumor cells, particularly glioma stem-like cells (GSCs), helps acquire this highly vascular phenotype by attracting blood vessels and inducing pro-angiogenic signaling in endothelial cells through extracellular matrix (ECM) remodeling (*Dittmer and Leyh, 2014*; *Rupp et al., 2016*). A reciprocal relationship exists between GSCs and endothelial cells by which endothelial cells induce stemness phenotype in cancer cells through activation of Notch, sonic hedgehog (SHH), and nitric oxide synthase signaling pathways (*Jeon et al., 2014*; *Yan et al., 2014*; *Zhu et al., 2011*), while GSCs drive vascularization of the tumor via endogenous endothelial cell stimulation, vascular mimicry, and GBM-endothelial cell transdifferentiation (*Hardee and Zagzag, 2012*; *Soda et al., 2011*). Recent reports have shown that CSCs induce high vascularization of tumors like GBM by migrating along blood vessel scaffolds to invade novel vascular niches, thereby ensuring surplus and continuous blood supply at their disposal (*Prager et al., 2020*). In GBM, CD133+ and Nestin+ cells (representing GSCs) are located in close proximity to the tumor microvascular density (MVD), whereas a lower number of CD133- and Nestin- cells (representing differentiated glioma cells [DGCs]) are located in the vicinity of the blood vessels. It has also been reported that the depletion of brain tumor blood vessels causes a decrease in the number of tumor-initiating GSCs (*Calabrese et al., 2007*).

While CD133 marker expression was reported to be associated with GSCs initially (*Singh et al., 2004*; *Galli et al., 2004*), later reports documented CD133- cells exhibiting GSC-like properties (*Beier et al., 2007*; *Chen et al., 2010*; *Joo et al., 2008*; *Ogden et al., 2008*; *Wang et al., 2008*). CXCR4-dependent SHH-GLI-NANOG signaling promotes stemness in GSCs. This study also showed that the miR302-367 cluster could suppress stemness and promote differentiation by targeting CXCR4/SDF1 (*Fareh et al., 2012*). The above group subsequently showed that miR18A* promotes GSC stemness by activating Notch-dependent SHH-GL1-NANOG signaling, targeting DLL3, an autocrine inhibitor of Notch 1 signaling (*Turchi et al., 2013*). In contrast to these observations, Dirkse et al. showed the existence of stem cell-associated heterogeneity in GBM, which results in tumor plasticity and is orchestrated by the microenvironment (*Dirkse et al., 2019*).

Besides CSC self-renewal, their differentiation to form the bulk cancer cells also plays a crucial role in tumor growth and maintenance (*Jin et al., 2017*). Epigenome unique to CSCs compared to differentiated cancer cells has been documented (*Suvà et al., 2014*; *Zhou et al., 2018*). Reciprocally, a set of four reprogramming transcription factors, POU3F2, SOX2, SALL2, and OLIG2, is identified in GBM that is sufficient to reprogram DGCs and create the epigenetic landscape of native GSCs, thus creating 'induced' CSCs (*Suvà et al., 2014*). The epigenetic regulation forms the basis of cellular plasticity, which creates a dynamic equilibrium between CSCs and differentiated cancer cells (*Safa et al., 2015*). Oncogene-induced dedifferentiation of mature cells in the brain was also reported using a mouse model of glioma, and the reprogrammed CSCs were proposed to contribute to the heterogeneous cell state populations observed in malignant gliomas (*Friedmann-Morvinski et al., 2012*; *Friedmann-Morvinski and Verma, 2014*). Lineage-tracing analyses revealed the reprogramming of DGCs to GSCs that act as a reservoir for initiating relapse of the tumors upon temozolomide chemotherapy (*Auffinger et al., 2014*; *Chen et al., 2012*). Hypoxia has also been reported to reprogram differentiated cells to form CSCs in glioma, hepatoma, and lung cancer (*Prasad et al., 2017*; *Wang et al., 2017*). Spontaneous conversion of differentiated cancer cells to CSCs has also been reported in breast cancer (*Klevebring et al., 2014*; *Zhou et al., 2019*).

Collectively, these studies highlight the crucial role of CSCs in cellular crosstalk in the tumor niche and establish CSCs as critical drivers of tumorigenesis. However, the massive imbalance in the

proportions of CSCs and non-CSCs or differentiated cancer cells in tumors raises several important questions. Considering that differentiated cancer cells constitute the bulk of tumors, do they have specific functions, or do they only constitute the tumor mass? Do they contribute to the complex paracrine signaling occurring within the tumor microenvironment? Do they support tumor growth by promoting CSC growth and maintenance? It has been recently shown in GBM that DGCs cooperate with GSCs through a paracrine feedback loop involving neurotrophin signaling to promote tumor growth (*Wang et al., 2018*). While this study suggests a supporting role for differentiated cancer cells in tumor growth, the large proportion of them in tumors suggests a role in paracrine interactions with other stromal cells in the tumor niche.

We used quantitative proteomics to identify DGC-secreted proteins that might support their paracrine interactions within the tumor microenvironment. We show an essential role of fibromodulin (FMOD) secreted by DGCs in promoting tumor angiogenesis via a crosstalk with endothelial cells. FMOD promotes integrin-dependent Notch signaling in endothelial cells to enhance their migratory and blood vessel-forming capacity. These findings indicate that DGCs are crucial for supporting tumor growth in the complex tumor microenvironment by promoting multifaceted interactions between tumor cells and the stroma.

## Results

### DGC and GSC secretomes have distinct proteomes revealed by tandem mass spectrometry

While GSCs alone can initiate a tumor, the overall tumor growth requires functional interactions between GSCs and DGCs (*Singh et al., 2004*; *Wang et al., 2018*). To further understand the respective roles of GSCs vs. DGCs in tumor growth, we compared the conditioned medium (CM) derived from three patient-derived human GSC cell lines (MGG4, MGG6, and MGG8) (*Wakimoto et al., 2009*) and their corresponding DGCs, using a quantitative proteomic strategy. Proteins in CMs were systematically analyzed by nano-flow liquid chromatography coupled to Fourier transform tandem mass spectrometry (nano-LC-FT-MS/MS), and their relative abundance in DGC vs. GSC CM was determined by label-free quantification. We found that 119 proteins are more abundant in GSC CM, while 185 proteins are more abundant in the DGC CM (p<0.05, *Figure 1A*; *Supplementary file 1*). Analysis of overrepresented functional categories among proteins exhibiting differential abundances in GSC vs. DGC CMs using Perseus with a p-value <0.05 revealed that the DGC CM is enriched in proteins known to exhibit extracellular or cell surface localization, such as proteins annotated as ECM organization while terms related to DNA replication and many signaling pathways are enriched in GSC CM (*Figure 1*; *Figure 1—figure supplement 1*).

### TGF-β signaling controls the expression of FMOD in DGCs

The enrichment of the ECM annotation among proteins exhibiting higher abundance in DGC secretome prompted us to focus on ECM proteoglycans in line with their critical role in facilitating cancer cell signaling through their interaction with growth factor receptors, extracellular ligands and matrix components, and in promoting tumor–microenvironment interactions (*Winkler et al., 2020*). Six ECM proteoglycans were found to be more abundant in DGC CM compared with GSC CM (*Figure 1B*). The role of five of them (LAMB2, SERPINEE1, ITGB1, TNC, and LAMA5) in tumor growth has been well established (*Angel et al., 2020*; *Bartolini et al., 2016*; *Long et al., 2016*; *Wang et al., 2021*; *Yang et al., 2020b*). We thus focused on FMOD, which exhibited the highest DGC CM/GSC CM protein ratio. FMOD is a small leucine-rich repeat proteoglycan upregulated in GBM due to the loss of promoter methylation orchestrated by TGF-β1-dependent epigenetic regulation (*Mondal et al., 2017*). FMOD promotes glioma cell migration through actin cytoskeleton remodeling mediated by an integrin-FAK-Src-Rho-ROCK signaling pathway but does not affect colony-forming ability, growth on soft agar, chemosensitivity, and glioma cell proliferation (*Mondal et al., 2017*). We first confirmed the higher abundance of FMOD seen in DGC CM compared to GSC CM (*Figure 1C*) both at the transcript level (*Figure 1D*) and at the protein level (*Figure 1E and F*) in three GSC cell lines (MGG4, MGG6, and MGG8).

In line with our previous findings indicating that TGF-β signaling controls FMOD expression in glioma (*Mondal et al., 2017*), we next explored the possible role of this pathway in FMOD overexpression

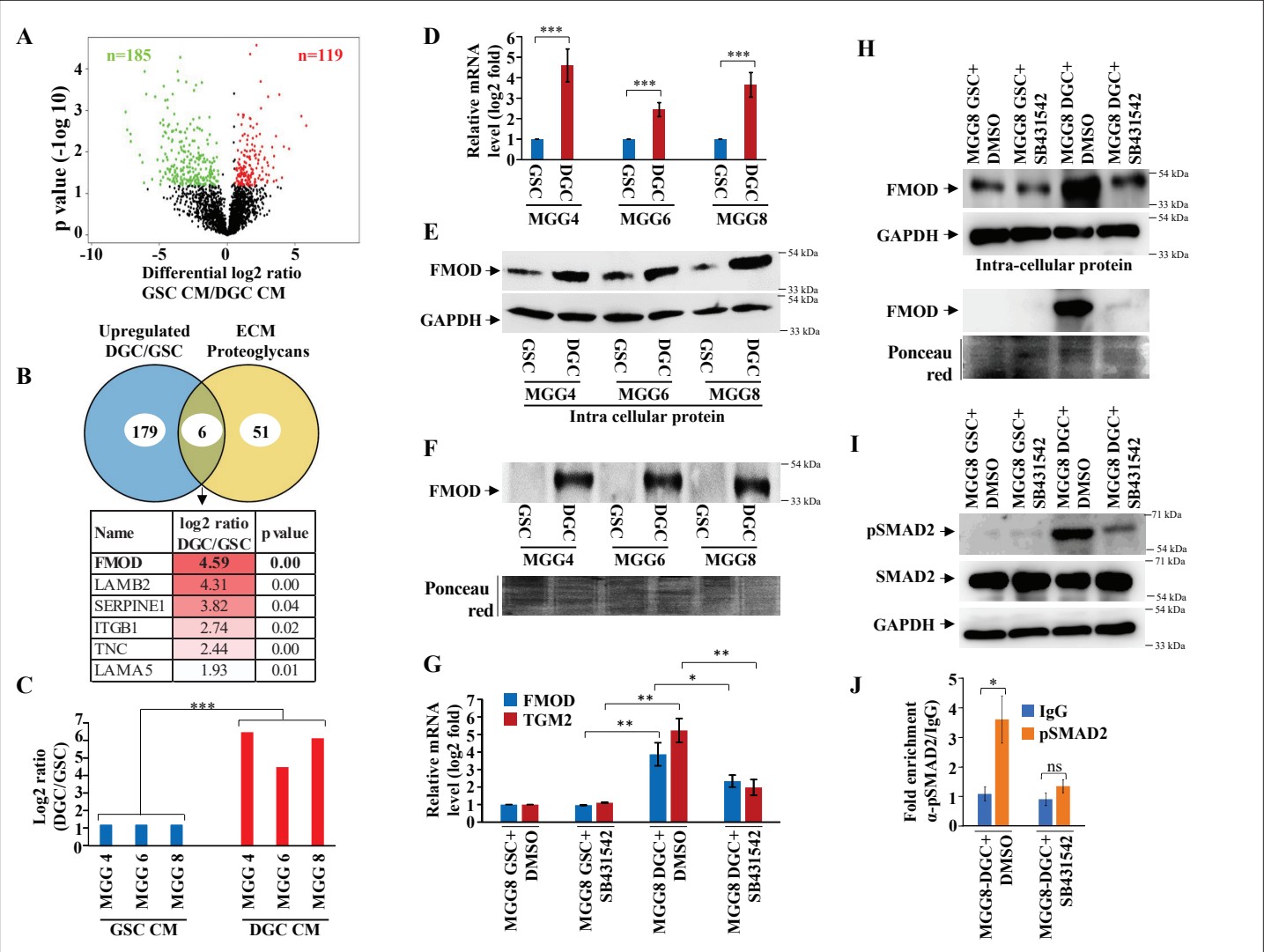

**Figure 1.** Quantitative proteomics shows a higher abundance of fibromodulin under the control of TGF-β signaling in the differentiated glioma cell (DGC) secretome. (**A**) Volcano plot depicting relative protein abundance in glioma stem-like cell (GSC) (MGG4, MGG6, and MGG8) vs. their corresponding DGC conditioned media (CM). The black dots represent the nonsignificant proteins (p>0.05), while the red (higher abundance in GSC CM) and green (lower abundance in GSC CM) dots represent the significant ones (p<0.05) with a log₂ fold change cutoff of >0.58 or <−0.58. (**B**) Venn diagram showing the relationship between proteins upregulated in DGC CM and annotated extracellular matrix (ECM) proteoglycans. Of the common proteins shown below, fibromodulin (FMOD) exhibits the highest DGC/GSC ratio (indicated by the more intense red color). (**C**) Label-free quantification (LFQ) of FMOD, expressed as log₂ fold change in GSCs vs. DGCs CM. (**D**) RT-qPCR analysis shows upregulation of FMOD transcript in DGCs (red bars) compared to GSCs (blue bars). (**E**) Western blotting shows the presence of higher amounts of intracellular FMOD in DGCs compared with corresponding GSCs. (**F**) Western blotting shows the presence of higher amounts of FMOD in the DGC CM compared to GSC CM (top panel). Equal loading of the proteins assessed by Ponceau Red staining (bottom panel). (**G**) RT-qPCR analysis shows a reduction of FMOD transcript level in DGCs, but not in GSCs, upon treatment with SB431542 (10 μM), a TGF-β inhibitor. Red bars indicate FMOD expression, and blue bars represent TGM2 (a bonafide TGF-β pathway target gene) expression. (**H**) Western blotting shows the reduction of FMOD protein level in DGCs, but not in GSCs, upon treatment with SB431542 (10 μM) (intracellular, top, and secreted, bottom). Equal loading of the secreted proteins assessed by Ponceau Red staining. (**I**) Western blotting shows higher expression of pSAMD2 in DGCs than in GSCs, which is reduced by SB431542 treatment. (**J**) RT-qPCR shows significantly higher fold enrichment of pSMAD2 in the FMOD promoter, which is inhibited upon SB431542 treatment (10 μM). For panels (**C**), (**D**), (**G**), and (**J**), n=3, and p-value is calculated by unpaired *t*-test with Welch's correction. p-Value <0.05 is considered significant with *, **, and *** representing p-values <0.05, 0.01, and 0.001, respectively.

The online version of this article includes the following source data and figure supplement(s) for figure 1:

**Source data 1.** Source data used to generate *Figure 1A*.

**Source data 2.** Source data used to generate *Figure 1B*.

**Source data 3.** Source data used to generate *Figure 1C*.

*Figure 1 continued on next page*

*Figure 1 continued*

**Source data 4.** Source data used to generate *Figure 1D*.

**Source data 5.** Source data used to generate *Figure 1E, F, H, I*.

**Source data 6.** Source data used to generate *Figure 1G*.

**Source data 7.** Source data used to generate *Figure 1J*.

**Source data 8.** TGF-β is activated in glioblastoma (GBM) over normal samples in multiple datasets.

**Source data 9.** Source data used to generate *Figure 1—source data 8*.

**Source data 10.** Mesenchymal gene expression signature and TGF-β signaling pathway are enriched in differentiated glioma cells (DGCs).

**Source data 11.** Source data used to generate *Figure 1—source data 10*.

**Figure supplement 1.** Gene Ontology (GO) analysis of differentially abundant proteins in glioma stem-like cell (GSC) and differentiated glioma cell (DGC) conditioned medium (CM).

**Figure supplement 1—source data 1.** Source data used to generate *Figure 1*, *Figure 1—figure supplement 1A B*.

**Figure supplement 2.** The TGF-β pathway is activated in differentiated glioma cells (DGCs) over glioma stem-like cells (GSCs).

**Figure supplement 2—source data 1.** Source data used to generate *Figure 1—figure supplement 2*.

**Figure supplement 3.** Fibromodulin (FMOD) expression and TGF-β signaling activation are significantly higher in mesenchymal glioblastoma (GBM) than in other subtypes.

**Figure supplement 3—source data 1.** Source data used to generate *Figure 1—figure supplement 3A–D*.

in DGCs. Gene set enrichment analysis (GSEA) of differentially regulated transcripts in GSC vs. DGC showed significant depletion of several TGF-β signaling pathway genes (*Figure 1—figure supplement 2*; *Supplementary file 2*), suggesting an enhanced TGF-β signaling in DGCs. Likewise, GSEA revealed an enrichment of several TGF-β signaling pathway genes in most GBM transcriptome datasets (*Figure 1—source data 8*), further supporting the activation of TGF-β signaling in DGCs that represent the bulk of GBMs. In addition, the DGCs used in this study that express a high level of FMOD showed enrichment in mesenchymal signature compared to GSCs (*Figure 1—source data 10*), consistent with the elevated TGF TGF-β signaling and FMOD levels we observed in the mesenchymal GBM subtype (*Figure 1—figure supplement 3B*). Moreover, treating MGG8-DGCs with the TGF-β inhibitor (SB431542) significantly decreased luciferase activity of SBE–Luc (a TGF-β-responsive reporter and contains Smad-binding elements) and FMOD Promoter-Luc reporters (*Figure 1—figure supplement 3C D*). We also found higher levels of FMOD and TGM2 (a bonafide TGF-β target gene) transcripts and FMOD and pSMAD2 (an indicator of activated TGF-β signaling) proteins in MGG8-DGCs than MGG8-GSCs (*Figure 1G–I*). The addition of a TGF-β inhibitor (SB431542) significantly decreased transcript levels of FMOD and TGM2 and protein levels of FMOD and pSMAD2 in MGG8-DGCs (*Figure 1G–I*). Further, chromatin immunoprecipitation experiments revealed pSMAD2 occupancy on FMOD promoter in MGG8 DGCs that was significantly reduced by pretreating cells with SB431542 (*Figure 1J*). These results demonstrate a predominant expression and secretion of FMOD by DGCs that are promoted by TGF-β signaling.

## Tumor growth requires FMOD secreted by DGCs

Toward exploring the possible role of DGC-secreted FMOD in glioma tumor growth, we first investigated the role of FMOD in GSC and DGC growth and interconversion between both cell populations in vitro using two human (MGG8 and U251) and two murine (AGR53 and DBT-Luc) glioma cell lines. We found that the absence of FMOD neither affected GSC growth and differentiation to DGC (*Figure 2—figure supplement 1*, *Figure 2—figure supplement 2*, *Figure 2—figure supplement 3*) nor DGC growth and reprogramming to form GSCs (*Figure 2—figure supplement 4*, *Figure 2—figure supplement 5*, *Figure 2—figure supplement 6*; more details in Appendix 1), consistent with our previous findings showing that FMOD does not affect glioma cell proliferation in vitro (*Mondal et al., 2017*). In line with previous findings that DGCs cooperate with GSCs to promote tumor growth (*Wang et al., 2018*), we then evaluated the ability of DGCs silenced for *FMOD* to support the growth of tumors initiated by GSCs in co-implantation experiments in a syngeneic mouse model using GSCs and DGCs derived from DBT-Luc glioma cells. Reminiscent of our observations in MGG8 cell line, DBT-Luc-DGCs express higher levels of *FMOD* than DBT-Luc-GSCs (*Figure 2—figure supplement 5C D*). To silence the expression of *FMOD* in DBT-Luc-DGCs,

we used a doxycycline-inducible construct that contains an inducible *mCherry-shRNA* downstream of the Tet-responsive element (*Angel et al., 2020*; *Figure 2A*). The scheme of the co-implantation experiment is described on *Figure 2B*. DBT-Luc-GSC cells were coinjected with either DBT-Luc-DGC/*miRNT* (nontargeting *shRNA*) or DBT-Luc-DGC/*miRFMOD* (*FMOD shRNA*). In both groups, 50% of the mice received doxycycline on alternated days from day 9 post-injection until the end of the experiment. Tumors in mice coinjected with DBT-Luc-GSCs and DBT-Luc-DGCs/*miRNT* grew much faster and reached a significantly larger size (measured by bioluminescence) than tumors in mice injected with DBT-Luc-GSCs alone regardless of doxycycline treatment (*Figure 2B–D*, compare black and purple lines with blue line; *Supplementary file 3*). Notably, mice treated with doxycycline did show mCherry expression in tumors (*Figure 2C*). In contrast, injected DBT-Luc-DGC/*miRFMOD* cells failed to support the growth of tumors initiated by DBT-Luc-GSCs in doxycycline-treated mice compared to doxycycline-untreated mice (*Figure 2B–D*, compare red line with orange line; *Supplementary file 3*). While mice injected with DBT-Luc-GSCs+DBT-Luc-DGCs/*miRFMOD* (Dox+) showed an increase in tumor growth until the onset of doxycycline treatment (as seen in the rise in bioluminescence), subsequent tumor growth was drastically reduced. As expected, mice injected with DBT-Luc-DGCs alone developed substantially small tumors (*Figure 2C and D*). The small tumors formed in animals injected with either DBT-Luc-GSC+DBT-Luc-DGC/*miRFMOD* (Dox+) or DBT-Luc-GSC alone expressed significantly less FMOD protein than other tumors (*Figure 2—figure supplement 7*). These results indicate that FMOD secreted by DGCs is essential for the growth of tumors initiated by GSCs.

## FMOD induces angiogenesis of host-derived and tumor-derived endothelial cells

Tumor cell interactions with stromal cells are critical for glioma tumor growth (*Pine et al., 2020*). Small leucine-rich proteoglycans such as FMOD promote angiogenesis in the context of cutaneous wound healing (*Pang et al., 2019*; *Zheng et al., 2014*). In addition, we previously found a significant enrichment of the term 'angiogenesis' among differentially regulated genes in FMOD-silenced U251 glioma cells (*Mondal et al., 2017*). In light of these observations, we next examined the impact of FMOD on tumor angiogenesis. First, we tested the ability of glioma cell-derived FMOD to induce angiogenic network formation by immortalized human pulmonary microvascular endothelial cells (ST1). We used LN229 and U251 glioma cells, which express low and high levels of FMOD, respectively, for overexpression and silencing studies (*Mondal et al., 2017*). We found that the CM derived from LN229 cells stably expressing FMOD (LN229/FMOD) induced more angiogenesis than LN229/Vector stable cells (*Figure 3A and B*). Further, the CM of FMOD-silenced U251 cells was less efficient in promoting angiogenesis than the CM of cells expressing nontargeting siRNA (*Figure 3C*, *Figure 3—figure supplement 1A*) or shRNA (*Figure 3—figure supplement 1B*). The addition of recombinant human FMOD (rhFMOD) to the CM of U251/siFMOD cells rescued its ability to induce angiogenesis (*Figure 3C*, *Figure 3—figure supplement 1A*). More importantly, the addition of rhFMOD directly to endothelial cells induced angiogenesis in the presence of a control antibody (IgG) but not in the presence of an FMOD neutralizing antibody (*Figure 3—figure supplement 1C*).

Both CMs derived from three DGCs and their corresponding GSCs also induced angiogenesis efficiently (*Figure 3—figure supplement 1D*). Further, pretreating cells with an FMOD antibody significantly reduced the ability of MGG8-DGC CM, but not that of MGG8-GSC CM, to induce angiogenesis (*Figure 3D*). The reduced ability of the FMOD antibody-pretreated DGC CM to promote angiogenesis was rescued by the exogenous addition of an excess of rhFMOD (*Figure 3D*). Moreover, CM derived from FMOD-silenced MGG8-DGCs was less efficient in promoting angiogenesis than the CM of shNT (*Figure 3D*) or siNT (*Figure 3—figure supplement 1E*) transfected MGG-DGCs. The effect of FMOD silencing was rescued by adding exogenous rhFMOD (*Figure 3D, Figure 3—figure supplement 1E*). Both rhFMOD and FMOD present in the CM collected from MGG8-DGC induced the migration and invasion but not the proliferation of ST1 cells (*Figure 3—figure supplement 2A–E*). Further, CM from MGG8-DGC/shNT cells was more efficient than CM from MGG8-DGC/shFMOD cells in promoting angiogenic network formation by human brain-derived primary endothelial cells (HBMECs) and mouse brain-derived immortalized endothelial cells (B.End3) (*Figure 3E and F*, *Figure 3—figure supplement 2F*). Again, the effect of FMOD silencing was rescued by adding rhFMOD (*Figure 3E and F*, *Figure 3—figure supplement 2F*).

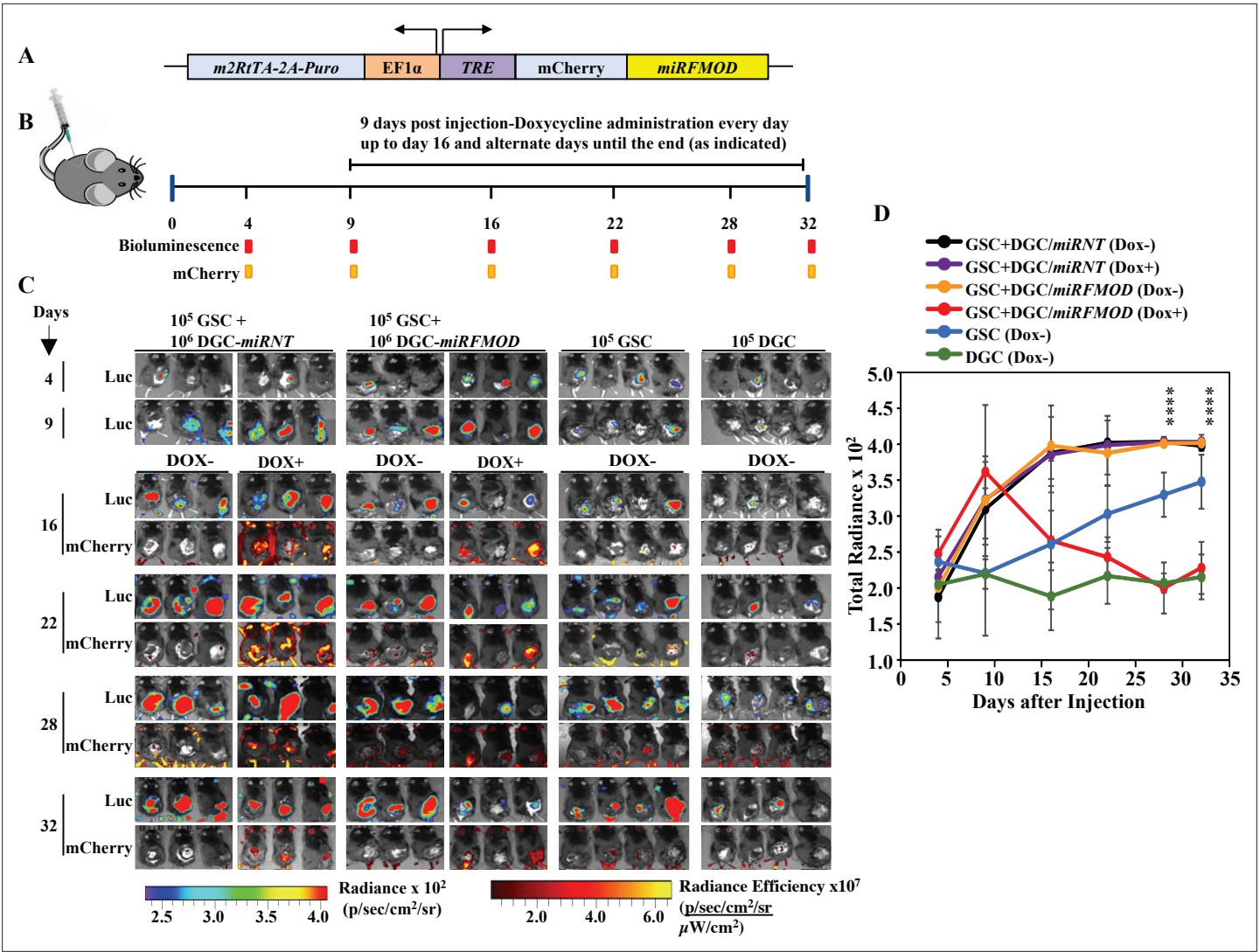

**Figure 2.** Differentiated glioma cell (DGC)-secreted fibromodulin (FMOD) is essential for tumor growth initiated by glioma stem-like cells (GSCs) in vivo in a co-implantation experiment. (**A**) Diagram of the inducible *shFMOD* lentiviral construct. (**B**) Schema depicts the GSC-DGC co-implantation experiment in C57BL/6 mice (n = 5 per group). Mice were injected subcutaneously with a combination of DBT-Luc-GSCs and DBT-Luc-DGCs transduced with either *miRNT* (nontargeting) or miRFMOD lentiviruses. To induce *miRNT* or *miRFMOD* (and *mCherry*), mice received doxycycline (100 µl of 1 mg/ml per animal) as intraperitoneal injections at indicated times. The control groups were only injected with DBT-Luc-GSCs or DBT-Luc-DGCs and did not receive doxycycline. (**C**) In vivo imaging of the injected mice shows tumor growth over time by bioluminescence and mCherry fluorescence, according to the timeline shown in (**B**). (**D**) Quantification of the total radiance. The different colors represent the different groups of animals. Significant differences between each of the groups were calculated using ANOVA. The p-values for days 28 and 32 are shown. A detailed comparison of the p-values between different groups is provided in ***Supplementary file 2***.

The online version of this article includes the following source data and figure supplement(s) for figure 2:

**Source data 1.** Source data used to generate ***Figure 2C***.

**Source data 2.** Source data used to generate ***Figure 2D***.

**Figure supplement 1.** Fibromodulin (FMOD) does not play a role in glioma stem-like cell (GSC) neurosphere formation/maintenance and differentiation in human GSCs.

**Figure supplement 1—source data 1.** Source data used to generate ***Figure 2—figure supplement 1A–E***.

**Figure supplement 2.** Difference in the level of fibromodulin (*FMOD*) expression between glioma stem-like cell (GSC) and differentiated glioma cell (DGC) of mouse glioma cell line AGR53 and confirmation of efficient conditional knockdown.

**Figure supplement 2—source data 1.** Source data used to generate ***Figure 2—figure supplement 2A–E***.

*Figure 2 continued on next page*

*Figure 2 continued*

**Figure supplement 3.** Fibromodulin (FMOD) does not play a role in glioma stem-like cell (GSC) neurosphere formation and differentiation in murine GSCs.

**Figure supplement 3—source data 1.** Source data used to generate *Figure 2—figure supplement 3A–D*.

**Figure supplement 4.** Fibromodulin (FMOD) does not play a role in glioma stem-like cell (GSC) differentiation and reprogramming of murine GSCs.

**Figure supplement 4—source data 1.** Source data used to generate *Figure 2—figure supplement 4A B*.

**Figure supplement 5.** Validation of fibromodulin (FMOD) levels and knockdown in DBT-Luc mouse glioma cell line.

**Figure supplement 5—source data 1.** Source data used to generate *Figure 2—figure supplement 5A–G*.

**Figure supplement 6.** Fibromodulin (FMOD) does not have a role in the de-differentiation of differentiated glioma cells (DGCs) to glioma stem-like cells (GSCs).

**Figure supplement 6—source data 1.** Source data used to generate *Figure 2—figure supplement 6A–D*.

**Figure supplement 7.** Induction with doxycycline reduces fibromodulin (FMOD) level in mice injected with DBT-GSC-Luc+DBT-DGC-Luc/*miRFMOD* cells.

**Figure supplement 7—source data 1.** Source data used to generate *Figure 2—figure supplement 7*.

Vascular mimicry (VM) is one of the alternative mechanisms of angiogenesis wherein tumor-derived endothelial cells (TDECs) originate from GBM cells (*Angara et al., 2017*; *Ricci-Vitiani et al., 2010*; *Soda et al., 2011*). To assess the ability of FMOD to induce TDECs derived from DGCs to form angiogenic networks, we used MGG8-DGC and U87 cells. MGG8-DGC/shNT cells grown in endothelial media (M199) under hypoxia (1% $O_2$) differentiated to TDECS as evidenced by an increase in CD31 (*Figure 3G and H*). MGG8-DGC/shFMOD also differentiated to form TDECs, albeit with less efficiency (*Figure 3G and H*). Further, the addition of rhFMOD induced both MGG8-DGC/shNT-TDEC and MGG8-DGC/shFMOD-TDEC cells to form angiogenic networks efficiently (*Figure 3I and J*). Similarly, U87 cells differentiated to TDECs (*Figure 3—figure supplement 3A B*), which readily formed angiogenic networks in the presence of rhFMOD (*Figure 3—figure supplement 3C D*). Collectively, these results demonstrate that DGC CM can induce angiogenesis and identify FMOD as a critical mediator of DGC-induced angiogenesis.

## FMOD activates integrin/FAK/Src-dependent Notch pathway in endothelial cells to induce angiogenesis

To dissect the signaling mechanisms underlying FMOD-induced angiogenesis, we subjected protein extracts derived from ST1 endothelial cells treated or not with rhFMOD to reverse phase protein array (RPPA). A total of 12 proteins exhibited differential abundance in a time-dependent manner in rhFMOD-treated ST1 endothelial cells (*Figure 4A*). These include HES1, a downstream target of the Notch signaling pathway that has been shown to promote angiogenesis (*Zhao et al., 2017*). We thus investigated the possible involvement of Notch signaling in FMOD-induced angiogenesis. The addition of rhFMOD induced luciferase activity of Notch-dependent CSL-Luc and HES-Luc reporters in ST1 cells but not in gamma-secretase inhibitor (GSI; a Notch pathway inhibitor) pretreated cells (*Figure 4—figure supplement 1A B*). The addition of rhFMOD also increased HES1 mRNA and protein levels in ST1 cells, an effect abolished by GSI pretreatment of cells (*Figure 4B and C*). rhFMOD treatment also resulted in the translocation of NICD (Notch intracellular domain) from the cytosol to the nucleus, as shown by subcellular fractionation and confocal microscopy (*Figure 4—figure supplement 1C D*). Furthermore, rhFMOD failed to induce angiogenic network formation by GSI-pretreated ST1 cells (*Figure 4D*). In addition, ST1 cells having a stable expression of NICD (ST1/NICD) showed enhanced angiogenic network formation than ST1 vector stable (ST1/Vector) cells (*Figure 4—figure supplement 2A–D*). FMOD present in CM from MGG8-DGCs induced ST1/Vector cells, but not ST1/NICD cells, to form more angiogenic networks, suggesting that Notch activation in endothelial cells is an essential step in FMOD-induced angiogenesis.

The increase in phosphorylated FAK (pFAK, FAK_Py397-R-V; the molecule downstream of integrin signaling) levels in rhFMOD-treated endothelial cells, as shown by RPPA (*Figure 4A*), also suggested a possible role of integrin signaling in FMOD-induced angiogenesis. This is consistent with our previous findings indicating that FMOD activates integrin signaling via type I collagen to engage the FAK-Src-Rho-ROCK pathway and promote the migration of glioma cells (*Mondal et al., 2017*). We first

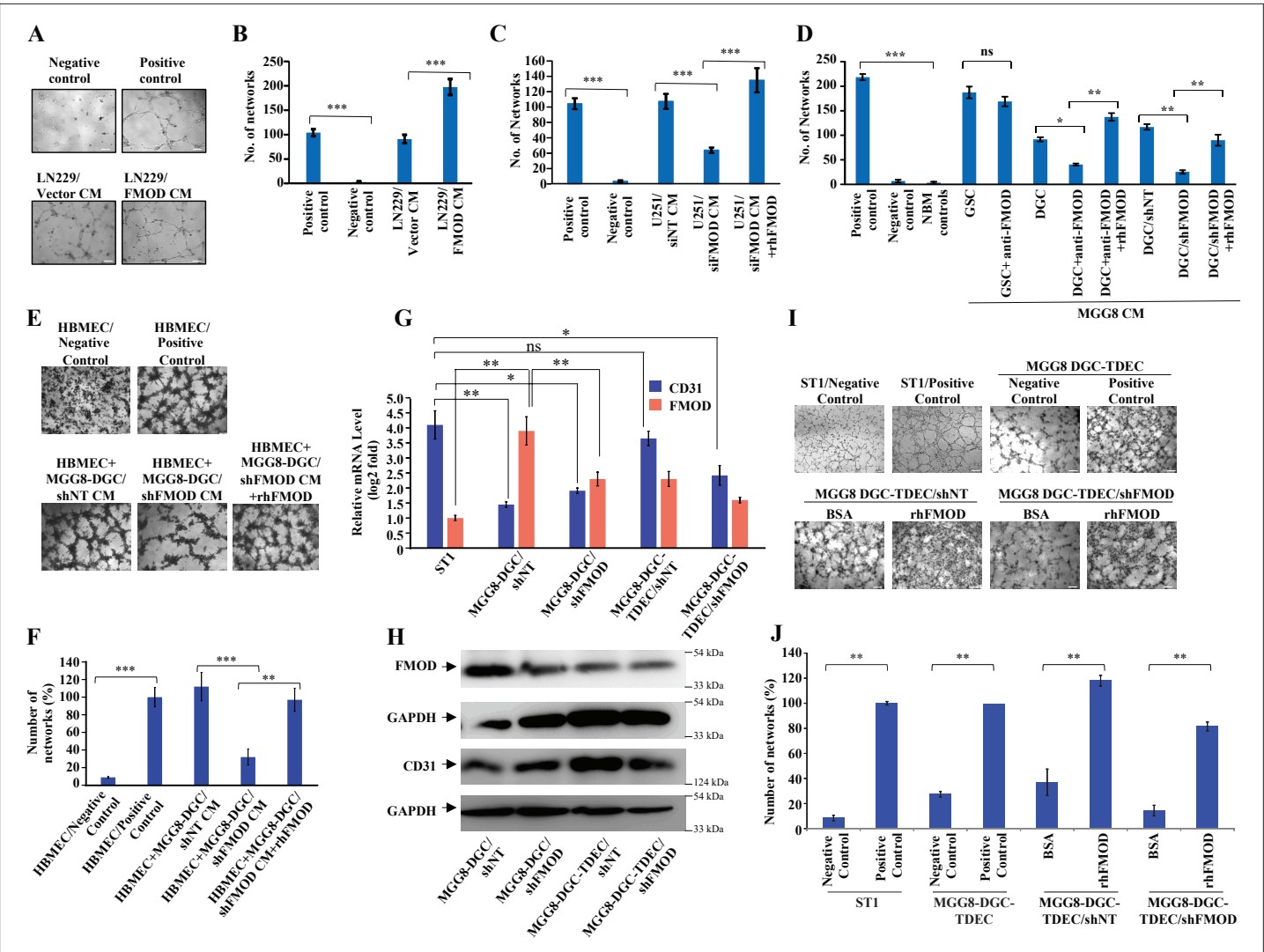

**Figure 3.** Differentiated glioma cell (DGC)-secreted fibromodulin (FMOD) promotes angiogenesis of host-derived and tumor-derived endothelial cells. (**A**) Representative images of in vitro network formation by ST1 cells treated with conditioned medium (CM) of LN229/Vector CM and LN229/FMOD. In the positive control condition (top right), cells are plated in complete endothelial cell media (M199) supplemented with endothelial cell growth factors (ECGS) and 20% fetal bovine serum (FBS), and in the negative control (top left), cells are plated in incomplete M199 (without serum and ECGS). Networks formed by ST1 cells treated with CM of LN229/Vector (left bottom) and LN229/FMOD (right bottom). Magnification = ×10, scale bar = 100 µm. (**B**) Quantification of the number of complete networks formed in (**A**). (**C**) Quantification of the number of networks formed by ST1 cells treated with CM of U251-DGC/siNT, U251-DGC/siFMOD, and U251-DGC/siFMOD + rhFMOD (400 nM) cells. (**D**) Quantification of the number of networks formed by ST1 cells treated with CM of MGG8-GSC, MGG8-DGC, MGG8-DGC/shNT, and MGG8-DGC/shFMOD supplemented with anti-FMOD or rhFMOD (400 nM) as indicated. (**E**) Representative images of in vitro network formation by primary human brain-derived microvascular endothelial cells (HBMECs). In the positive control condition (top right), HBMEC cells are plated in complete endothelial cell media (M199) supplemented with ECGS and 20% FBS, and in the negative control (top left), cells are plated in incomplete M199 without serum and ECGS. Networks formed by HBMEC cells treated with CM of MGG8-DGC/shNT, MGG8-DGC/shFMOD, and MGG8-DGC/shFMOD + rhFMOD (400 nM). Magnification = ×4, scale bar = 200 µm. (**F**) Quantification of the number of complete networks formed in (**E**). (**G**) RT-qPCR analysis showing transcript levels of CD31 (blue bars) and FMOD (orange bars) in ST1, MGG8-DGC/shNT, MGG8-DGC/shFMOD, MGG8-DGC-TDEC/shNT, and MGG8-DGC-TDEC/shFMOD cells. (**H**) Western blotting shows FMOD and CD31 protein levels in MGG8-DGC/shNT, MGG8-DGC/shFMOD, MGG8-DGC-TDEC/shNT, and MGG8-DGC-TDEC/shFMOD cells. (**I**) Representative images of in vitro network formation by ST1, MGG8-DGC-TDEC/shNT, and MGG8-DGC-TDEC/shFMOD upon bovine serum albumin (BSA) and rhFMOD (400 nM) treatments. Top panels: in the positive control conditions, ST1 or MGG8-DGC-TDEC cells are plated in complete endothelial cell media (M199) supplemented with ECGS and 20% FBS, and in the negative control conditions, ST1 or MGG8-DGC-TDEC cells are plated in incomplete M199 (without serum and ECGS). Bottom panels: networks formed by HBMEC cells treated with CM of MGG8-DGC-TDEC/shNT and MGG8-DGC-TDEC/shFMOD supplemented with either BSA or rhFMOD (400 nM). Magnification = ×4, scale bar = 200 µm. (**J**) Quantification of the number of complete networks formed in (**I**). For panels (**B–D**), (**F**), (**G**), and (**J**), n=3 and p-values were calculated by unpaired t-test with Welch's correction. p-Value <0.05 was considered significant with *, **, and *** representing p-values <0.05, 0.01, and 0.001, respectively. ns, nonsignificance.

*Figure 3 continued on next page*

*Figure 3 continued*

The online version of this article includes the following source data and figure supplement(s) for figure 3:

**Source data 1.** Source data used to generate *Figure 3A*.

**Source data 2.** Source data used to generate *Figure 3B*.

**Source data 3.** Source data used to generate *Figure 3C*.

**Source data 4.** Source data used to generate *Figure 3D*.

**Source data 5.** Source data used to generate *Figure 3E*.

**Source data 6.** Source data used to generate *Figure 3F*.

**Source data 7.** Source data used to generate *Figure 3G*.

**Source data 8.** Source data used to generate *Figure 3H*.

**Source data 9.** Source data used to generate *Figure 3*.

**Source data 10.** Source data used to generate *Figure 3J*.

**Figure supplement 1.** Differentiated glioma cell (DGC)-secreted fibromodulin (FMOD) induces angiogenesis by endothelial cells of various origins.

**Figure supplement 1—source data 1.** Source data used to generate *Figure 3—figure supplement 1A–E* .

**Figure supplement 2.** Fibromodulin (FMOD) induces migration and invasion, but not the proliferation of endothelial cells.

**Figure supplement 2—source data 1.** Source data to generate *Figure 3—figure supplement 2A–F*.

**Figure supplement 3.** Fibromodulin (FMOD) induces glioblastoma (GBM) cells to undergo transdifferentiation.

**Figure supplement 3—source data 1.** Source data used to generate *Figure 3—figure supplement 3A–D*.

confirmed the activation of integrin signaling by FMOD, as assessed by increased pFAK in rhFMOD-treated ST1 cells, but not in cells pretreated with RGD peptide, an integrin inhibitor (*Figure 4E*). The addition of RGD peptide inhibited angiogenesis induced by LN229/FMOD CM (*Figure 4F*). Likewise, angiogenesis induced by LN229/FMOD CM was completely inhibited when ST1 cells were pretreated with inhibitors of FAK (FAKi) or Src (PP2), two signaling molecules downstream of integrin (*Figure 4F*). These treatments also strongly reduced the basal level of angiogenesis elicited by the CM of LN229/Vector cells. In contrast, an inactive analog of Src inhibitor (PP3), as well as inhibitors of RAC1 and ROCK, failed to inhibit the ability of CM derived from LN229/FMOD cells to induce angiogenesis (*Figure 4F*). Our previous report also demonstrated that the interaction of FMOD with type I collagen is essential for integrin activation (*Mondal et al., 2017*). The C-terminal region of FMOD comprises 11 leucine-rich-repeats (LRRs), of which the 11th repeat binds to type I collagen (*Oldberg et al., 2007*). A synthetic interfering peptide (RLDGNEIKR) corresponding to the 11th[h] LRR of type 1 collagen, but not a modified peptide (RLDGNQIMR), competes with rhFMOD for binding to type I collagen to activate integrin signaling in glioma cells (*Mondal et al., 2017*; *Oldberg et al., 2007*). Consistent with these findings, rhFMOD-induced luciferase activity of CSL-Luc and HES-Luc (*Figure 4—figure supplement 3A B*) and angiogenesis by ST1 cells (*Figure 4—figure supplement 3C*) were significantly inhibited by the interfering peptide, but not the modified peptide, suggesting a crucial role of type I collagen-dependent activation of integrin signaling in FMOD-induced angiogenesis. To identify the α and β subunits of integrin involved in FMOD-mediated activation of integrin signaling in endothelial cells, we chose ITGA6, ITGB1, and ITGAV for investigation based on the analysis of transcriptome data derived from laser capture-dissected microvessels from the human brain (more details in Appendix 1). Silencing either of the selected three integrin subunits in ST1 cells reduced significantly the ability of rhFMOD to activate integrin as assessed by reduced pFAK levels (*Figure 4—figure supplement 3D–I*), thus demonstrating the involvement of αv/β1 and α6/β1 heterodimeric integrin receptors in FMOD activation of integrin signaling in endothelial cells.

Next, to examine a possible crosstalk between integrin and Notch signaling in FMOD-treated endothelial cells, we tested the effect of the RGD peptide on the ability of rhFMOD to induce Notch signaling. Pretreatment of ST1 cells with RGD peptide significantly reduced rhFMOD-elicited CSL-Luc and HES-Luc activation (*Figure 4—figure supplement 1E F*). Likewise, pretreatment of cells with FAKi or PP2, but not PP3, significantly reduced rhFMOD-induced CSL-Luc and HES-Luc activity in ST1 cells (*Figure 4—figure supplement 4A–D*). Further, rhFMOD failed to increase HES1 transcript and protein levels in ST1 cells treated with either RGD peptide, FAKi, or PP2, but not in cells treated with PP3 (*Figure 4G and H*, *Figure 4—figure supplement 4E–H*).

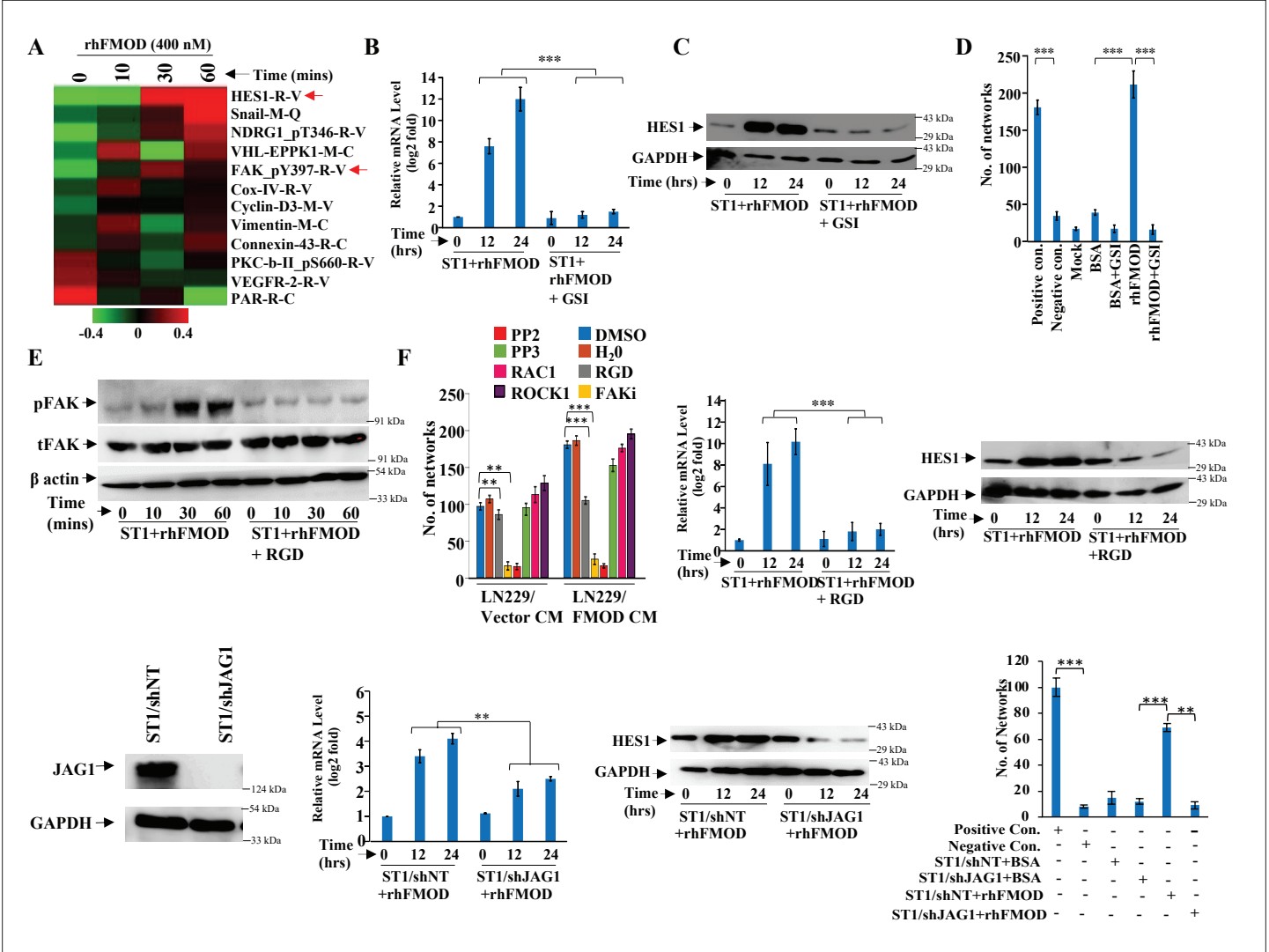

**Figure 4.** Integrin/FAK/Src/JAG1-dependent Notch pathway activation in endothelial cells mediates fibromodulin (FMOD)-induced angiogenesis. (**A**) Heatmap showing differentially regulated ($\log_2$ fold change >/<0.2) proteins in ST1 cells treated with vehicle or rhFMOD (400 nM) for 10, 30, and 60 min, assessed by reverse phase protein array (RPPA). Red and green depict upregulated and downregulated proteins, respectively. The red arrows indicate HES1 and pFAK proteins. (**B**) RT-qPCR analysis shows HES1 transcript levels in ST1 cells treated with rhFMOD (400 nM) with or without gamma-secretase inhibitor (GSI; 10 µM). (**C**) Western blotting shows HES1 protein levels in ST1 cells treated with rhFMOD (400 nM) with or without GSI (10 µM). (**D**) Quantification of the number of networks formed by ST1 cells treated with rhFMOD (400 nM) with or without GSI (10 µM). (**E**) Western blotting shows phospho-FAK levels in ST1 cells treated with rhFMOD (400 nM) with or without RGD peptide (10 µM). (**F**) Quantification of the number of networks formed by ST1 cells pretreated with indicated small-molecule inhibitors (PP2 [10 µM], PP3 [10 µM], and PF573228 [FAK inhibitor; 10 µM], H1152 [ROCK1 inhibitor; 0.5 mM], and Rac1 inhibitor [10 µM]) followed by incubation with conditioned medium (CM) of LN229/Vector or LN229/FMOD cells. (**G**) RT-qPCR analysis shows HES1 transcript levels in ST1 cells treated with rhFMOD (400 nM) with or without RGD peptide (10 µM). (**H**) Western blotting shows HES1 protein levels in ST1 cells treated with rhFMOD (400 nM) with or without RGD peptide (10 µM). (**I**) Western blotting shows JAG1 protein levels in ST1/shNT and ST1/shJAG1 cells. (**J**) RT-qPCR analysis shows HES1 transcript levels in ST1/shNT and ST1/shJAG1 cells treated with rhFMOD (400 nM). (**K**) Western blotting shows HES1 protein levels in ST1/shNT and ST1/shJAG1 cells treated with rhFMOD (400 nM). (**L**) Quantification of the number of networks formed by ST1/shNT and ST1/shJAG1 cells treated with bovine serum albumin (BSA) or rhFMOD (400 nM). For panels (**B**), (**D**), (**F**), (**G**), (**J**), and (**L**), n=3, and the p-values were calculated by unpaired *t*-test with Welch's correction are indicated. p-Value <0.05 was considered significant with *, **, and *** representing *p*-values <0.05, 0.01, and 0.001, respectively.

The online version of this article includes the following source data and figure supplement(s) for figure 4:

**Source data 1.** Source data used to generate *Figure 4A*.

**Source data 2.** Source data used to generate *Figure 4B*.

**Source data 3.** Source data used to generate *Figure 4C, E, H, I*.

*Figure 4 continued on next page*

*Figure 4 continued*

**Source data 4.** Source data used to generate *Figure 4D*.

**Source data 5.** Source data used to generate *Figure 4F*.

**Source data 6.** Source data used to generate *Figure 4G*.

**Source data 7.** Source data used to generate *Figure 4J*.

**Source data 8.** Source data used to generate *Figure 4K*.

**Source data 9.** Source data used to generate *Figure 4L*.

**Figure supplement 1.** Fibromodulin (FMOD) induces activation of Notch signaling in endothelial cells.

**Figure supplement 1—source data 1.** Source data used to generate *Figure 4—figure supplement 1A–F*.

**Figure supplement 2.** ST1 cells stably expressing NICD are independent of fibromodulin (FMOD) in forming angiogenic networks.

**Figure supplement 2—source data 1.** Source data used to generate *Figure 4—figure supplement 2A–D*.

**Figure supplement 3.** Fibromodulin (FMOD)–type I collagen interaction is crucial for FMOD-mediated activation of downstream signaling pathways.

**Figure supplement 3—source data 1.** Source data used to generate *Figure 4—figure supplement 3A–I*.

**Figure supplement 4.** Fibromodulin (FMOD)-mediated activation of Integrin-dependent Notch signaling in endothelial cells involves the integrin pathway downstream molecules FAK and Src.

**Figure supplement 4—source data 1.** Source data used to generate *Figure 4—figure supplement 4A–H*.

**Figure supplement 5.** Fibromodulin (FMOD)-mediated crosstalk of integrin and Notch signaling pathways occurs via JAG1 upregulation.

**Figure supplement 5—source data 1.** Source data used to generate *Figure 4—figure supplement 5A–H*.

**Figure supplement 6.** Clinical relevance of the FMOD-JAG1-HES1 signaling.

**Figure supplement 6—source data 1.** Source data used to generate *Figure 4—figure supplement 6A–E*.

**Figure supplement 7.** Clinical relevance of the FMOD-JAG1-HES1 signaling.

**Figure supplement 7—source data 1.** Source data used to generate *Figure 4—figure supplement 7A B*.

We next investigated the mechanistic link between integrin and Notch signaling in FMOD-treated ST1 cells. Since Notch activation by FMOD is sensitive to GSI treatment, we explored the activation of Notch ligands by integrin-FAK signaling in FMOD-treated cells. While the addition of rhFMOD induced DLL3 and JAG1 transcripts in ST1 cells, RGD peptide pretreatment abolished JAG1 induction (*Figure 4—figure supplement 5A*), suggesting that JAG1 might be a potential linking molecule. Consistently, pretreatment of cells with either FAKi or PP2, but not PP3, also abolished the ability of rhFMOD to induce JAG1 transcript in ST1 cells (*Figure 4—figure supplement 5B C*). Further, JAG1 silencing significantly decreased rhFMOD-induced activity of CSL-Luc and HES-Luc reporters and HES1 transcript/protein levels in ST1 cells (*Figure 4—figure supplement 5D E*, *Figure 4I–K*). JAG1 silencing also abolished the ability of rhFMOD to induce angiogenic networks by ST1 cells (*Figure 4L*). In addition, rhFMOD induced the transcript levels of KLF8, a FAK-inducible transcription factor in ST1 cells, but not in RGD peptide pretreated cells (*Figure 4—figure supplement 5F*). Further, KLF8 silencing in ST1 cells significantly reduced the ability of rhFMOD to increase JAG1 level (*Figure 4—figure supplement 5G H*), suggesting that KLF8 activates JAG1 through an integrin-dependent pathway in FMOD-treated endothelial cells. Collectively, these results identify JAG1 as a molecular link between integrin and Notch signaling pathways in FMOD-treated endothelial cells and demonstrate a key role of integrin-FAK-JAG1-Notch-HES1 signaling in FMOD-induced angiogenesis.

To further explore the clinical relevance of these findings, we interrogated transcriptome datasets from multiple sources (more details in Appendix 1). We found a significant upregulation of transcript levels of FMOD, JAG1, and HES1 in GBM from various datasets (*Figure 4—figure supplement 6A–C*). We also found a positive correlation between FMOD and HES1 transcripts and between FMOD and JAG1 transcripts in the majority of GBM datasets analyzed (*Figure 4—figure supplement 6D E*), which further substantiates the functional link between FMOD, JAG1, and HES1. We also show that high FMOD transcript levels and hypomethylation of FMOD promoter are associated with poor prognosis in most datasets (*Figure 4—figure supplement 7A B*). These observations provide additional support for the activation of integrin-Notch signaling in FMOD-treated endothelial cells.

## DGC-secreted FMOD is required for the growth of murine and human GSC-initiated tumors

While GSCs alone can initiate a tumor, tumor growth requires continuous differentiation to form DGCs, which form the bulk of the tumor mass. In line with our co-implantation experiments (*Figure 2*), we sought to define the importance of FMOD secreted by DGCs generated through a differentiation program initiated by GSCs during tumor growth in vivo using a syngenic intracranial glioma mouse model. We injected AGR53-GSC-*miRNT* and AGR53-GSC-*miRFMOD* cells intracranially into C57/black mice and allowed them to form tumors. 13 days after intracranial injections, both groups received doxycycline as indicated (*Figure 5A*). The understanding is that as the tumors start growing, GSCs, in addition to their self-renewal, will start differentiating de novo to form DGCs, which would express high levels of *FMOD*. However, doxycycline treatment would inhibit *FMOD* expression, and thus one could investigate the importance of DGC-secreted FMOD in tumor growth. AGR53-GSC/*miRNT* and AGR53-GSC/*miRFMOD* cell-initiated tumors showed a similar size 7 days after doxycycline treatment as shown by mCherry fluorescence (*Figure 5B and C*; day 21). However, upon subsequent follow-up, doxycycline administration significantly inhibited the growth of AGR53-GSC/*miRFMOD*-initiated tumors over time but not that of AGR53-GSC/*miRNT* tumors (*Figure 5B, C and E*). Further, doxycycline administration increased the survival of mice injected with AGR53-GSC/*miRFMOD* cells compared to AGR53-GSC/*miRNT* cells (*Figure 5D*). AGR53-GSC/*miRFMOD*-initiated tumors showed decreased FMOD, mCherry, and GFP expression compared to AGR53-GSC/*miRNT*-initiated tumors (*Figure 5F and G*). DBT-Luc GSC/*miRFMOD*, another murine glioma GSC cell line, produced similar results: FMOD silencing after doxycycline administration resulted in reduced tumor growth (*Figure 5—figure supplement 1A B*), increased mice survival (*Figure 5—figure supplement 1C*), and decreased FMOD expression (*Figure 5—figure supplement 1D*).

To determine the relevance of our findings to the human pathology, we investigated the importance of DGC-secreted FMOD in the growth of tumors initiated by MGG8 and U251 cells using a xenograft mouse glioma model. The higher expression of CD133 confirmed the enrichment of CSCs in MGG8-GSC neurospheres compared to MGG8-DGCs (*Figure 6A*). We established orthotopic xenografts using MGG8-GSC/shNT and MGG8-GSC/shFMOD cells. Reminiscent of results obtained using transplantation of murine glioma cells, MGG8-GSC/shNT-transplanted mice readily developed intracranial tumors, whereas MGG8-GSC/shFMOD mice showed impaired tumor formation and increased mice survival (*Figure 6B and C*). Immunostaining and confocal microscopy analysis showed high expression of FMOD in MGG8-GSC/shNT tumors, while FMOD was barely detectable in MGG8-GSC/shFMOD tumors (*Figure 6D*). Likewise, U251/shFMOD cells that showed reduced FMOD protein levels compared to U251/shNT (*Figure 2—figure supplement 6A*) developed smaller tumors, and mice bearing U251/shFMOD tumors had longer survival than those carrying U251/shNT tumors (*Figure 6—figure supplement 1A–D*). As expected, the expression of FMOD was more elevated in U251/shNT tumors than in U251/shFMOD tumors (*Figure 6—figure supplement 1E*). Collectively, these findings indicate that DGC-secreted FMOD is essential for the growth of both human and mouse glioma.

## Reduced angiogenesis in small tumors formed in FMOD-silenced conditions

Next, we investigated the cellular differentiation and angiogenesis in the tumors toward understanding the mechanism underlying reduced tumor growth in FMOD-silenced conditions. First, we measured the expression of CD133 and GFAP markers as the representation of GSCs and DGCs in the tumors formed in the animal models. The CD133-positive cells are much less in proportion compared to GFAP-positive cells in tumors formed by AGR53-GSC, DBT-Luc-GSCs, and MGG8-GSCs under FMOD nontargeting conditions (*Figure 7—figure supplement 1A B*, *Figure 7—figure supplement 2A B*, *Figure 7—figure supplement 3A B*), in good correlation to the low proportion of GSCs seen in brain tumors (*Singh et al., 2004*; *Galli et al., 2004*; *Calabrese et al., 2007*). Further, most GFAP-positive cells are also positive for FMOD expression compared to CD133-positive cells (*Figure 7—figure supplement 1A C*, *Figure 7—figure supplement 2A C*, *Figure 7—figure supplement 3A C*), recapitulating the results we obtained in vitro, where FMOD is expressed specifically by DGCs (*Figure 1*). We also found a similar expression pattern of CD133 and GFAP markers in small tumors formed under FMOD-silenced conditions in all three tumor models (*Figure 7—figure supplement 1D E*,

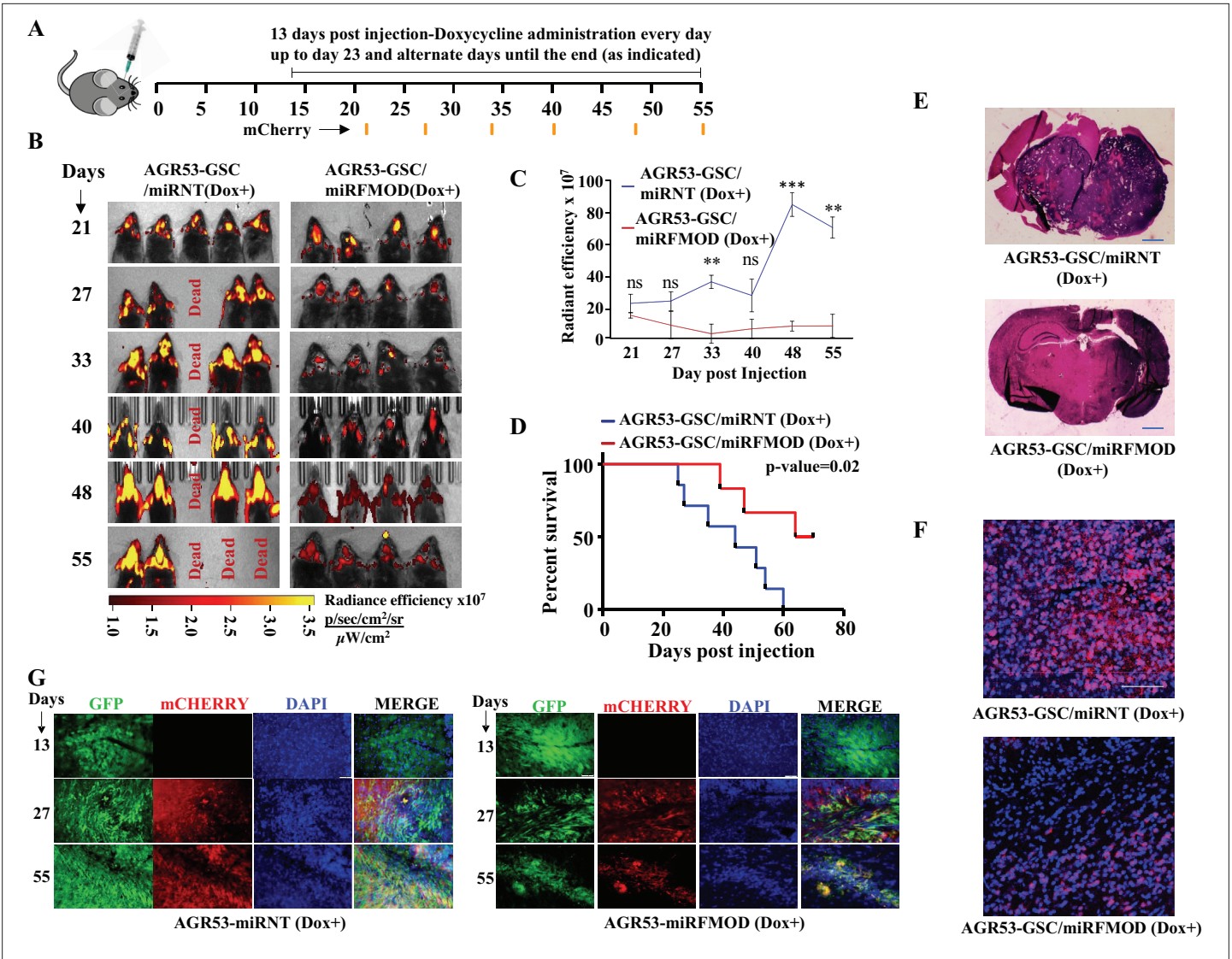

**Figure 5.** Conditional silencing of fibromodulin (*FMOD*) in differentiated glioma cells (DGCs) formed de novo by glioma stem-like cell (GSC)-initiated tumors inhibits tumor growth. (**A**) Schema depicts the timeline of the intracranial orthotopic mouse glioma model using murine glioma stem cells (AGR53-GSC) in C57BL/6 mice (n = 10 per group). Mice were injected with AGR53-GSCs (1 × 10⁵ cells per animal) transduced with either *miRNT* (nontargeting) or *miRFMOD* lentiviruses. Tumors were allowed to grow till day 13 and then *miRNT* or *miRFMOD* (and *mCherry*) were induced by doxycycline (100 μl of 1 mg/ml per animal) intraperitoneal injections at indicated times. Note that in vitro characterization shows that the highest knockdown of *FMOD* was obtained on the seventh day after doxycycline administration. First, in vivo imaging for mCherry expression depicting tumor size was done on day 21 post-injection, followed by imaging at regular intervals (as noted by the orange marks). (**B**) In vivo fluorescence (mCherry) imaging of mice injected with either AGR53-GSC/*miRNT* or AGR53-GSC/*miRFMOD* cells as per the timeline shown in (**A**). (**C**) The radiance efficiency for each time point in the two groups of animals as indicated was plotted. (**D**) Kaplan–Meier graph showing the survival of mice bearing tumors formed by AGR53-GSC/*miRNT* (Dox+) and AGR53-GSC/*miRFMOD* (Dox+) cells. (**E**) Hematoxylin and eosin staining shows a larger tumor (depicted by dark blue color due to extremely high cell density) in mice brain injected with AGR53-GSC/*miRNT* (Dox+) cells (top) compared to that of AGR53-GSC/*miRFMOD* (Dox+) cells (bottom). Magnification = ×0.8. (**F**) Confocal microscopy analysis showing FMOD expression in brains of mice injected with AGR53-GSC/*miRNT* (Dox+) and AGR53-GSC/*miRFMOD* (Dox+) cells. Red indicates FMOD, and blue indicates H33342 (stains nuclei). The merged images are shown for representation. Magnification = ×20, scale bar = 50 μm. (**G**) Brain sections showing areas of fluorescence (GFP, mCherry, and DAPI) for both AGR53-GSC/*miRNT* (Dox+) (left panel) and AGR53-GSC/*miRFMOD* (Dox+) (right panel) groups of animals. Note that the AGR53 cell line stably expresses GFP while mCherry expression is induced upon doxycycline addition. On day 13, prior to the administration of doxycycline, both AGR53-GSC/*miRNT* (left) and AGR53-GSC/*miRFMOD* (right) do not have any mCherry expression but have almost similar GFP expression. However, over time after the onset of doxycycline administration, both mCherry and GFP expression decreased in the *miRFMOD* group but not in the *miRNT* group. Merged images show an overlap of GFP and mCherry-positive tumor areas. Magnification = ×20, scale bar = 50 μm. For panel (**C**), the p-value was

*Figure 5 continued on next page*

*Figure 5 continued*

calculated by unpaired *t*-test with Welch's correction, and for (**D**), the p-value was calculated by log-rank test. p-Value <0.05 was considered significant with *, **, and *** representing p-values <0.05, 0.01, and 0.001, respectively.

The online version of this article includes the following source data and figure supplement(s) for figure 5:

**Source data 1.** Source data used to generate *Figure 5B*.

**Source data 2.** Source data used to generate *Figure 5C*.

**Source data 3.** Source data used to generate *Figure 5D*.

**Source data 4.** Source data used to generate *Figure 5E*.

**Source data 5.** Source data used to generate *Figure 5F*.

**Source data 6.** Source data used to generate *Figure 5G*.

**Figure supplement 1.** Conditional silencing of fibromodulin (*FMOD)* in differentiated glioma cells (DGCs) formed de novo by glioma stem-like cell (GSC)-initiated tumors inhibits tumor growth in a murine glioma model.

**Figure supplement 1—source data 1.** Source data used to generate *Figure 5—figure supplement 1*.

*Figure 7—figure supplement 2D E*, *Figure 7—figure supplement 3D E*). The GFAP staining in these tumors confirms the occurrence of an efficient differentiation program even in FMOD-silenced conditions, which confirms our results obtained in vitro, where the absence of FMOD failed to affect the GSC differentiation to form DGCs (*Figure 2—figure supplement 1*, *Figure 2—figure supplement 2*, *Figure 2—figure supplement 3*). As expected, the small tumors formed in FMOD-silenced conditions showed substantially reduced FMOD staining (*Figure 7—figure supplement 1D F*, *Figure 7—figure supplement 2D F*, *Figure 7—figure supplement 3D F*).

We next evaluated the extent of blood vessel formation by measuring the endothelial cell marker immunostaining in the tumors. The small size tumors formed by AGR53-GSC/*miRFMOD* cells after doxycycline treatment showed reduced staining for CD31 and von Willebrand factor (vWF) compared to AGR53-GSC/*miRNT* tumors (*Figure 7A-C*, *Figure 7—figure supplement 4A–C*). Tumors formed by DBT-Luc-GSC/*miRFMOD* cells in doxycycline-treated mice also showed significantly reduced CD31 staining compared to that measured in the absence of doxycycline treatment (*Figure 7—figure supplement 4D–F*). Reminiscent of murine glioma tumors, tumors induced by MGG8-GSC/shFMOD cells also showed reduced CD31 staining compared to MGG8-GSC/shNT cells (*Figure 7D–F*). We then tested the extent of blood vessel formation by TDECs. In all three tumor models (AGR53-GSCs, DBT-Luc-GSCs, and MGG8-GSCs), blood vessels formed by TDECs were significantly reduced in tumors formed in FMOD-silenced conditions (*Figure 7G–L*, *Figure 7—figure supplement 4G–I*). These findings confirm our previous results obtained in vitro, where the absence of FMOD decreased the ability of host-derived endothelial cells and TDECs to form angiogenic networks (*Figure 3—figure supplement 1*, *Figure 3—figure supplement 2*, *Figure 3—figure supplement 3*). Next, we investigated the involvement of integrin-FAK-JAG1-Notch-HES1 signaling in FMOD-induced angiogenesis in the context of glioma tumors. Confocal microscopy analysis in tumors formed by AGR53-GSCs, DBT-Luc-GSCs, and MGG8-GSCs revealed a significant colocalization of the endothelial cell marker CD31 with FMOD, pFAK, JAG1, and HES1 markers in blood vessels (*Figure 7—figure supplement 5A–C*). From these results, we conclude that angiogenesis induced by DGC-secreted FMOD is essential for glioma tumor growth.

## Discussion

The importance of GSCs in tumor initiation, growth, immune escape, angiogenesis, invasion into the normal brain, and resistance to therapy is also well established (*Bao et al., 2006*; *Wakimoto et al., 2009*). GSCs are known for their ability to self-renew and differentiate to form DGCs, the bulk cells of tumors (*Galli et al., 2004*; *Ignatova et al., 2002*; *Singh et al., 2004*; *Suvà et al., 2014*; *Yang et al., 2020a*). However, the role of DGCs in tumor growth remains poorly understood. The key requirement of tumor cells for self-maintenance in a novel tumor niche is the supply of nutrients. GSCs are known to promote the establishment of a highly vascularized microenvironment by being in close physical contact with endothelial cells (*Calabrese et al., 2007*). In addition, GSC-secreted factors such as VEGF-A, GDF15, IL8, and miR21 also have been shown to induce tumor angiogenesis (*Conroy et al., 2018*; *Sun et al., 2017*; *Thirant et al., 2013*; *Treps et al., 2017*; *Zhu et al., 2021*).

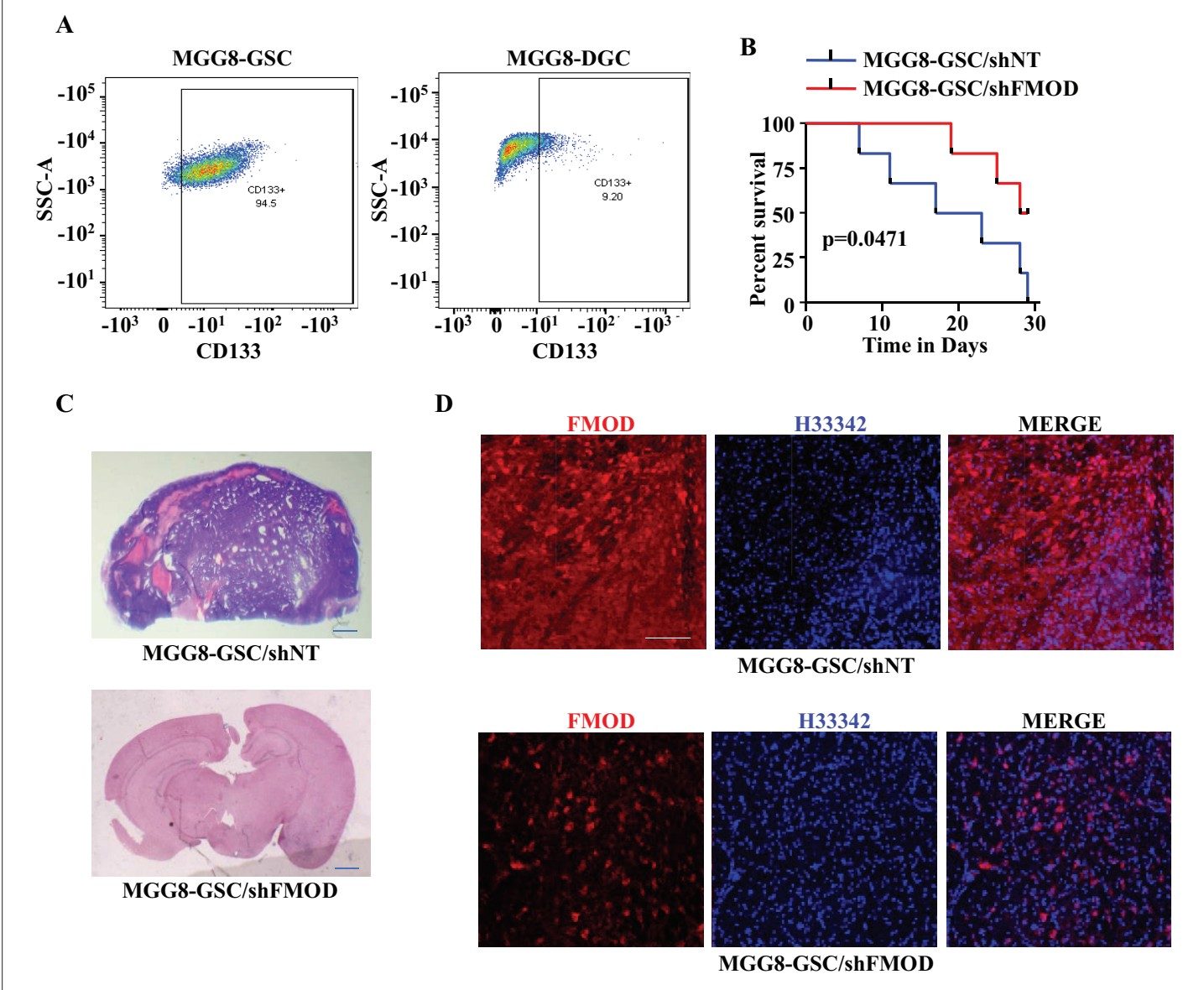

**Figure 6.** Growth of human glioma stem-like cell (GSC)-initiated tumors requires secreted fibromodulin (FMOD). (**A**) Flow cytometry analysis shows the relative levels of CD133-positive cells in MGG8-GSCs compared to MGG8-DGCs. (**B**) Kaplan–Meier graph shows the survival of mice (n = 10 per group) injected intracranially with MGG8-GSC/shNT or MGG8-GSC/shFMOD cells (1 × 10⁵ cells per animal). (**C**) Hematoxylin and eosin staining shows a larger tumor (depicted by dark blue color due to high cell density) in mice brain injected with MGG8-GSC/shNT cells (top) compared to that of MGG8-GSC/ shFMOD cells (bottom). Magnification = ×0.8. (**D**) Confocal microscopy analysis shows FMOD expression in brains of mice injected with MGG8-GSC/ shNT (top panel) and MGG8-GSC/shFMOD (bottom panel) cells. Red indicates FMOD, and blue indicates H33342 (stains nuclei). Magnification = ×20, scale bar = 50 µm. Statistical significance for panel (**B**) was calculated using the log-rank test. p-Value <0.05 is considered significant.

The online version of this article includes the following source data and figure supplement(s) for figure 6:

**Source data 1.** Source data used to generate *Figure 6A*.

**Source data 2.** Source data used to generate *Figure 6B*.

**Source data 3.** Source data used to generate *Figure 6C*.

**Source data 4.** Source data used to generate *Figure 6D*.

**Figure supplement 1.** Growth of human glioblastoma (GBM) cell-initiated tumors requires secreted fibromodulin (FMOD).

**Figure supplement 1—source data 1.** Source data used to generate *Figure 6—figure supplement 1*.

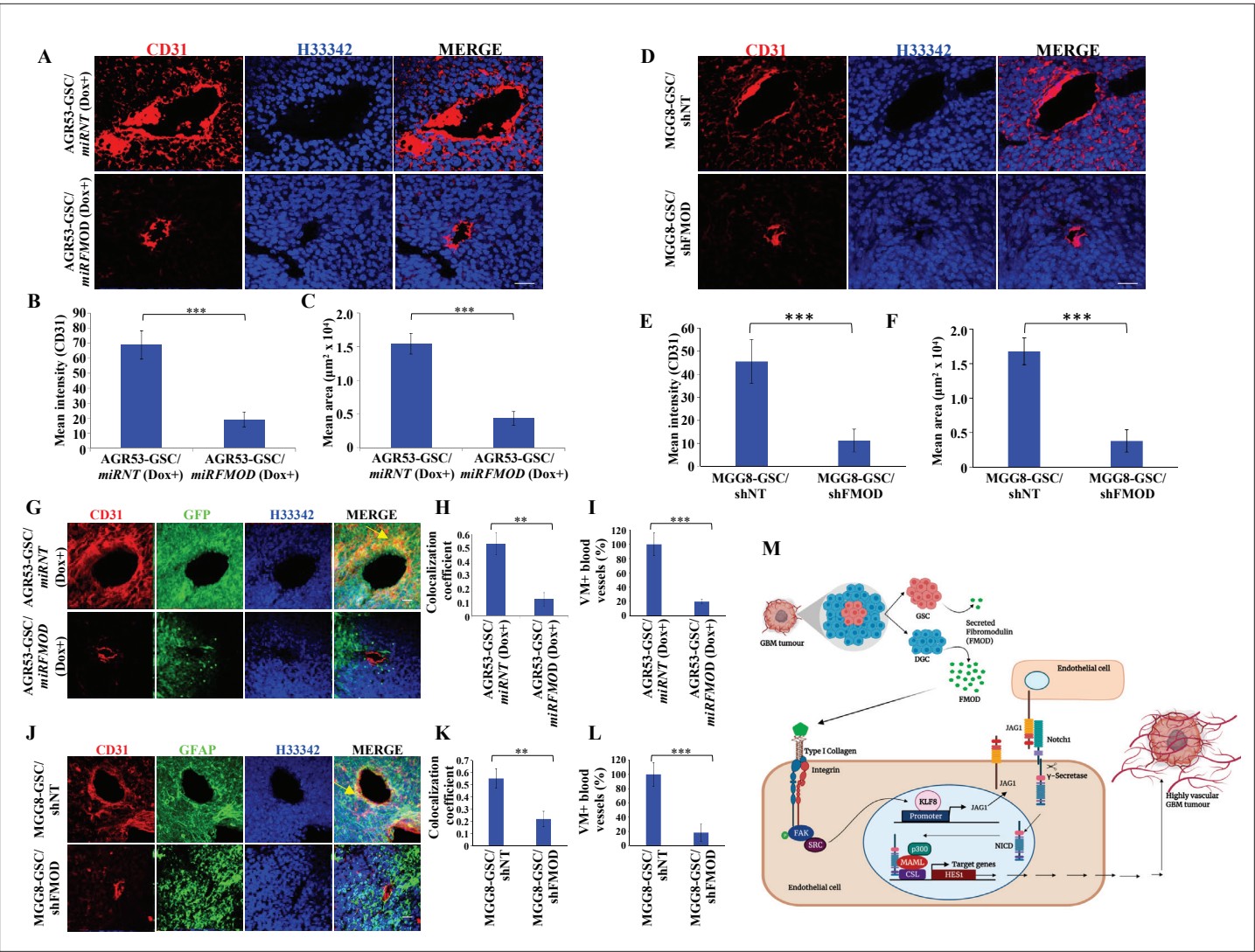

**Figure 7.** Reduced angiogenesis is characteristic of tumors initiated by fibromodulin (FMOD)-silenced glioma cells. (**A**) Confocal microscopy analysis shows CD31 expression in brain tumor sections of mice injected with AGR53-GSC/*miRNT* or AGR53-GSC/*miRFMOD* cells after doxycycline administration to the mice. Red indicates CD31, and blue indicates H33342 (stains nuclei). Magnification = ×20, scale bar = 50 μm.(**B**) Quantification of the mean fluorescence intensity of CD31 in brain tumor sections of mice injected with AGR53-GSC/*miRNT* or AGR53-GSC/*miRFMOD* cells after doxycycline administration to the mice. (**C**) Quantification of the mean area of blood vessels in brain tumor sections of mice injected with AGR53-GSC/*miRNT* or AGR53-GSC/*miRFMOD* cells after doxycycline administration to the mice. (**D**) Confocal microscopy analysis showing CD31 expression in brain tumor sections of mice injected with MGG8-GSC/shNT or MGG8-GSC/shFMOD cells. Red indicates CD31, and blue indicates H33342 (stains nuclei). Magnification = ×20, scale bar = 50 μm. (**E**) Quantification of the mean fluorescence intensity of CD31 in brain tumor sections of mice injected with MGG8-GSC/shNT or MGG8-GSC/shFMOD cells. (**F**) Quantification of the mean area of blood vessels in brain tumor sections of mice injected with MGG8-GSC/shNT or MGG8-GSC/shFMOD cells. (**G**) Confocal microscopy analysis showing CD31 and GFP expression in brain tumor sections of mice injected with AGR53-GSC/*miRNT* or AGR53-GSC/*miRFMOD* cells after doxycycline administration to the mice. Red indicates CD31, green indicates GFP, and blue indicates H33342 (stains nuclei). The yellow arrow indicates the region exhibiting colocalization of CD31 and GFP. Magnification = ×20, scale bar = 50 μm. (**H**) Quantification of the colocalization coefficient of CD31 and GFP staining in the brain tumor sections of mice injected with AGR53-GSC/*miRNT* and AGR53-GSC/*miRFMOD* cells after doxycycline injection to the mice. (**I**) The number of blood vessels with co-staining of CD31 and GFP was quantified in brain tumor sections of mice injected with AGR53-GSC/*miRNT* and AGR53-GSC/*miRFMOD* cells after doxycycline injection to the mice and plotted. (**J**) Confocal microscopy analysis showing CD31 and GFAP expression in brain tumor sections of mice injected with MGG8-GSC/shNT or MGG8-GSC/shFMOD cells. Red indicates CD31, green indicates GFAP, and blue indicates H33342 (stains nuclei). The yellow arrow indicates the region exhibiting colocalization of CD31 and GFAP. Magnification = ×20, scale bar = 50 μm. (**K**) Quantification of the colocalization coefficient of CD31 and GFAP staining in the brain tumor sections of mice injected with MGG8-GSC/shNT and MGG8-GSC/shFMOD cells. (**L**) The number of blood vessels with co-staining of CD31 and GFAP was quantified in brain tumor sections of mice injected with MGG8-GSC/shNT and MGG8-GSC/shFMOD cells and plotted. (**M**) A model depicting the functional interactions between different cell types in glioblastoma (GBM) tumors, which comprise a small proportion of glioma stem-like cells (GSCs), a massive number of differentiated glioma cells (DGCs), and stromal cells. FMOD, primarily secreted by

*Figure 7 continued on next page*

*Figure 7 continued*

DGCs, upregulates JAG1 through the activation of integrin signaling in endothelial cells. The higher expression of JAG1 causes the activation of the Notch signaling pathway, which results in the transcriptional upregulation of HES1 in endothelial cells. The integrin-dependent Notch pathway activation promotes angiogenesis and vascular mimicry, leading to glioma tumor growth. For panels (**B**), (**C**), (**E**), (**F**), (**H**), (**I**), (**K**), and (**L**), n=5, and p-value was calculated using an unpaired *t*-test with Welch's correction. p-Value <0.05 was considered significant, with *, **, and *** representing p-values <0.05, 0.01, and 0.001, respectively.

The online version of this article includes the following source data and figure supplement(s) for figure 7:

**Source data 1.** Source data used to generate *Figure 7A*.

**Source data 2.** Source data used to generate *Figure 7B, C, H, I*.

**Source data 3.** Source data used to generate *Figure 7D*.

**Source data 4.** Source data used to generate *Figure 7E, F, K, L*.

**Source data 5.** Source data used to generate *Figure 7G*.

**Source data 6.** Source data used to generate *Figure 7J*.

**Source data 7.** Source data used to generate *Figure 7M*.

**Figure supplement 1.** Fibromodulin (*FMOD*) silencing does not hamper the differentiation potential of tumor cells in murine glioma models.

**Figure supplement 1—source data 1.** Source data used to generate *Figure 7—figure supplement 1A–F*.

**Figure supplement 2.** Fibromodulin (*FMOD*) silencing does not hamper the differentiation potential of tumor cells in murine glioma models.

**Figure supplement 2—source data 1.** Source data used to generate *Figure 7—figure supplement 2A–F*.

**Figure supplement 3.** Fibromodulin (*FMOD*) silencing does not hamper the differentiation potential of tumor cells.

**Figure supplement 3—source data 1.** Source data used to generate *Figure 7—figure supplement 3A–F* .

**Figure supplement 4.** Fibromodulin (*FMOD*) silencing affects angiogenesis and vascular mimicry in vivo.

**Figure supplement 4—source data 1.** Source data used to generate *Figure 7—figure supplement 4A–I*.

**Figure supplement 5.** Fibromodulin (FMOD)-dependent activation of the integrin-FAK-JAG1-HES1 signaling axis is maintained in vivo.

**Figure supplement 5—source data 1.** Source data used to generate *Figure 7—figure supplement 5*.

The massive proportion of DGCs in the tumor suggests that GSC-initiated angiogenesis might not be sufficient to meet the large nutrient requirement of the entire tumor. Wang et al. demonstrated that DGC-secreted BDNF is essential for GSC growth and maintenance through DGC-GSC paracrine signaling, which highlights the crucial role of DGC-secreted proteins in tumor formation (*Wang et al., 2018*). Further, a possible interaction between DGCs and stromal cells, such as endothelial cells, cannot be ruled out. We hypothesized that in addition to GSCs, DGCs might play an essential role in autocrine and paracrine signaling involving different types of cells to augment tumor growth. This study demonstrates the existence of paracrine signaling between DGCs and endothelial cells, which promotes angiogenesis and glioma tumor growth.

Previously, we have demonstrated that FMOD is highly expressed in GBMs compared to normal brain tissues. The loss of FMOD expression hampers the migratory function of glioma cells but has no impact on glioma cell proliferation (*Mondal et al., 2017*). This study identifies that FMOD is expressed exclusively by DGCs and further showed that FMOD is not needed for GSC/DGC growth and their plasticity to form one from the other. Given these facts, the inability of DGCs silenced for FMOD to support the growth of tumors initiated by GSCs was unexpected. However, it enlightened us that DGC-secreted FMOD has some essential yet unidentified functions supporting GSC-initiated tumor growth.

Here, we explored the mechanisms underlying the role of DGCs acting as critical support for GSC-initiated tumor growth. We demonstrate that DGC-secreted FMOD promotes tumor growth by inducing angiogenesis through integrin-dependent Notch signaling in endothelial cells, thus highlighting the importance of DGCs in tumor–stroma interactions that contribute to a sustainable niche for tumor growth. We further investigated the mechanism by which FMOD activates integrin-dependent Notch signaling in endothelial cells. Based on RPPA data that showed upregulation of HES1 in endothelial cells upon FMOD treatment, we demonstrated that activation of Notch signaling is essential for FMOD-induced angiogenesis. The importance of Notch signaling in glioma, especially in GSC growth and angiogenesis, is well documented (*Bazzoni and Bentivegna, 2019*; *Stockhausen et al., 2010*; *Teodorczyk and Schmidt, 2014*). Similarly, activation of Notch signaling in endothelial

cells has been involved in tumor development and angiogenesis (*Gridley, 2007*; *Kofler et al., 2011*; *Phng and Gerhardt, 2009*). RPPA data also showed an increase in pFAK in FMOD-treated glioma cells. Our previous study showed that FMOD acts on glioma cells via the integrin-FAK-Src-Rho axis to promote migration (*Mondal et al., 2017*). This study demonstrates that FMOD-activated integrin signaling is essential for Notch pathway activation in ST1 cells. Integrin signaling in endothelial cells has been shown to play a crucial role in angiogenesis (*Short et al., 1998*; *Silva et al., 2008*). Based on our results indicating that FMOD-activated Notch signaling in endothelial cells is inhibited by GSI, we predicted that FMOD-elicited activation of the Notch pathway could involve Notch ligand-dependent process. Our experiments identified that JAG1 is the linker molecule that integrates integrin signaling to the Notch pathway. We also found a significant colocalization of endothelial cell marker CD31 with FMOD, pFAK, JAG1, and HES1 in blood vessels of tumors formed in mouse models. Finally, endothelial cells stably expressing NICD showed enhanced angiogenic network formation in an FMOD-independent manner, suggesting that Notch activation is an essential step in FMOD-induced angiogenesis. Thus, our results show that integrin-FAK-Src-KLF8-JAG1-dependent Notch signaling activation in endothelial cells mediates FMOD-induced angiogenesis.

Finally, in an orthotopic intracranial GBM mouse model, we show that conditional silencing of *FMOD* in newly generated DGCs during tumor growth leads to a significant reduction of tumor growth. Supporting our in vitro data indicating that FMOD is not required for GSC growth and differentiation, we found that the small tumors formed in FMOD-silenced conditions show differentiation of GSCs. However, these tumors exhibited poorly developed blood vasculatures of host-derived endothelial cells and TDECs. The lower tumor burden in the absence of FMOD might be attributed to insufficient nutrient supply to sustain tumor growth due to the reduced blood vessel density. Thus, our study establishes an essential role of paracrine signaling between the DGCs and the stroma in the context of tumor growth in the natural tumor niche complexity. It also demonstrates the importance of FMOD secreted by DGCs in promoting human glioma tumor growth in a mouse model. We propose a tumor evolution model (*Figure 7M*), whereby GSCs, in addition to their self-renewal, continuously differentiate to form DGCs, which secrete protein factors like FMOD that mediate paracrine signaling in the different cell types of the tumor, thus creating a niche favorable to tumor growth.

In conclusion, this study demonstrates that in addition to GSCs, DGCs have an essential role in tumor growth and maintenance. While the therapy-resistant and self-renewing GSCs trigger the early events of transformation and growth, DGCs, the proportion of which continues to increase during tumor growth, progressively become essential. Thus, targeting both CSCs and differentiated cancer bulk cells is vital to achieving a durable therapeutic response. The study also highlights the potential of GSC and DGC CM analysis to uncover novel targets in cancer therapy and the critical influence of DGC-secreted FMOD in glioma tumor growth.

## Materials and methods
### Resource availability
Further information and requests for resources and reagents should be directed to and will be fulfilled by the lead contact Dr. Kumaravel Somasundaram (skumar1@iisc.ac.in).

### Materials availability
This study did not generate any new unique reagents.

### Experimental model and subject details
Experiments were performed in C57&BL/6J female mice and Athymic Nude female mice (6–8 weeks old) following the approval by the Institute Ethical Committee for Animal Experimentation (Institute Animal Ethics Committee [IAEC] Project Number: CAF/Ethics/752/2020). The mice were kept in a 12 hr light and dark cycle, fed ad libitum with a normal diet, and the experiments were done in the light phase of the cycle.

### Cell lines used
Primary human tumor-derived GSCs MGG4, MGG6, and MGG8 were extensively characterized and kindly gifted to us by Dr. Wakimoto, Massachusetts General Hospital, Boston, USA (*Wakimoto et al.,*

*2009*) . Characterization of AGR53 mouse-derived cell line has been described before (*Angel et al., 2020*; *Friedmann-Morvinski et al., 2016*). DBT-Luc cells were extensively characterized (*Jiang et al., 2014*) and gifted to us by Dr. Dinesh Thotala, Washington University in St. Louis, St Louis, MO. ST1 endothelial cells were extensively characterized (*Krump-Konvalinkova et al., 2001*) and gifted to us Dr. Ron Unger, Johannes Gutenberg University, Germany. B.End3 cell line was purchased from The American Type Culture Collection (ATCC, #CRL-2299, Certification of Analysis can be downloaded from https://www.atcc.org/products/crl-2299 with the lot# 70030469). The HBMECs were purchased from Cell Biologics, USA (#H-6023). The Certificate of Analysis can be found at https://cellbiologics.com/index.php?route=product/product&product_id=2959. The U251 cell line was bought from the European Collection of Authenticated Cell Cultures (ECACC, # 09063001). The Certificate of Analysis can be found at https://www.culturecollections.org.uk/products/celllines/generalcell/detail.jsp?refId=09063001&collection=ecacc_gc. The U87 cell line was bought from the ECACC (#89081402). The Certificate of Analysis can be found at https://www.culturecollections.org.uk/products/celllines/generalcell/detail.jsp?refId=89081402&collection=ecacc_gc. The LN229 cell line was bought from ATCC (# CRL-2611, https://www.atcc.org/products/crl-2611). All cell lines were obtained from the sources mentioned above within a year of experimentation, and additional yearly authentications were carried out wherever necessary. All cell lines are verified to be mycoplasma free by EZdetect PCR Kit for Mycoplasma Detection (HiMedia).

## Plasmids

The RAR3G vector (in which *miRNT* and *miRFMOD* are cloned for the inducible *shRNA* experiments) was previously described (*Friedmann-Morvinski et al., 2012*). *shRNA* for *FMOD* (*miRFMOD*) was cloned following the *mir30* (*miRNT*)-inducible backbone under the *TRE* (tetracycline response element) promoter, which is placed downstream of the *mCherry* reporter gene. The *m2RtTA* transactivator is expressed from the same vector in the opposite direction under the EF1α promoter, and following a 2A peptide, the puromycin gene is placed for selection in vitro. The FMOD overexpression plasmid was bought from Origene, USA (#LY419579). NICD pCMV Neo/intracellular domain of human Notch1 (NIC-1) and CSL Luc were kind gifts from Prof. Thomas Kandesch, Department of Genetics, University of Pennsylvania School of Medicine. Hes-Luc plasmid was bought from Addgene, USA (#43806).

## Antibodies used

### Primary antibodies

FMOD (Abgent AP9243b, 1:2000), FAK (Cell Signaling Technology 3285S, 1:1000), pFAK (Cell Signaling Technology #3283S, 1:500), GAPDH (Sigma #G8795, 1:20,000), Actin (Sigma A3854, 1:20,000), HES1 (Cell Signaling Technology #D6P2U, 11988S, 1:1000), vWF (Abcam #6994, 1:2000), CD31 (Cell Signaling Technology #89C2, mouse mAb 1:200 for IHC), JAG1 (Cell Signaling Technology, Jagged1 [D4Y1R] XP rabbit mAb #70109, 1:200 for IHC, 1:1000 for WB), FMOD (fibromodulin polyclonal antibody PA5-26250, Invitrogen, IHC 1:100, WB 1:1000), CD133 (recombinant anti-CD133 antibody-EPR20980-104 #ab216323 Abcam, Flow Cyt. 1:100, Prom1 monoclonal antibody-2F8C5, #MA1-219, IHC 1:100), SOX2 (Cell Signaling Technology #3579 rabbit mAb, IHC 1:100), GFAP (anti-GFAP antibody ab7260, Abcam ICC 1:200, recombinant anti-GFAP antibody [EPR1034Y] – mouse IgG2a [ab279290], IHC 1:200), NICD (NOTCH1 (cleaved Val1744) polyclonal antibody PA5-99448, WB 1:200), SMAD2 (Smad2 [D43B4] XP rabbit mAb #5339, WB 1:1000), pSMAD2 (phospho-SMAD2 [Ser465/Ser467] [E8F3R] rabbit mAb #18338, WB 1:1000), integrin beta-1 (Cell Signaling Technology #9699 rabbit mAb, WB 1:1000), integrin alphaV (Cell Signaling Technology #4711 rabbit Ab, WB 1:1000), integrin alpha-6 (recombinant anti-integrin alpha 6 antibody [EPR18124] [ab181551], WB 1:500), and KLF8 (anti-KLF8 antibody [ab168527], WB 1:500).

### Secondary antibodies

Goat anti-mouse HRP conjugate (Bio-Rad #170-5047, WB 1:5000), goat anti-rabbit (H+L) secondary HRP conjugate (Invitrogen, #31460, WB 1:5000), goat anti-mouse IgG (H+L) highly cross-adsorbed secondary antibody, Alexa Fluor 488 (Invitrogen, #A-11029), goat anti-rabbit IgG (H+L) highly cross-adsorbed secondary antibody, Alexa Fluor 488 (Invitrogen, Cat# A-11034), goat anti-mouse IgG (H+L) highly cross-adsorbed secondary antibody, Alexa Fluor 594 (Invitrogen, Cat# A-11032), goat anti-rabbit IgG (H+L) highly cross-adsorbed secondary antibody, Alexa Fluor 594 (Invitrogen, Cat#

A-11037), goat anti-rabbit IgG (H+L) highly cross-adsorbed secondary antibody, Alexa Fluor 405 Plus (Invitrogen, #A48254), and goat anti-mouse IgG (H+L) highly cross-adsorbed secondary antibody, Alexa Fluor 405 Plus (Invitrogen, Cat# A48255). All Alexa Fluor conjugated antibodies were used at a dilution of 1:500 for IHC and ICC.

## Recombinant protein
Fibromodulin (FMOD) (NM_002023) Human, Origene, Cat# TP306534.

## Neurosphere culturing
The GSCs were obtained by dissecting GBM tumor tissue and then treating with trypsin, followed by a trypsin inhibitor. The chopped tissue was then passed through a cell strainer to remove the debris. The obtained filtrate was plated in ultra-low attachment plates using the stem cell for neurosphere formation media described next. Then, neurospheres were grown in Neurobasal medium (#21103049, Gibco) supplemented with 1× L-glutamine (#25030081, stock 200 mM, i.e., 100×, Gibco), heparin 2 µg/ml (#H3393 Sigma), 1× B27 supplement (#17504044, Gibco, stock concentration 50×), 0.5× N2 supplement (#17502048, Gibco, stock concentration 100×), 20 ng/ml rhEGF (#g5021, Promega), 2 ng/ml rhFGF-basic (#100-18B-100 µg, PeproTech), and penicillin and streptomycin. To make single-cell suspensions for re-plating, the spheres were chemically dissociated after 7 days of plating, using NeuroCult Chemical Dissociation Kit (mouse, #05707) from Stem Cell Technologies, according to the manufacturer's instructions.

## Differentiation of GSCs to DGCs
For differentiation of GSCs to DGCs, fully grown neurospheres were collected from the ultra-low attachment plates (Corning, USA) and plated on normal cell culture dishes in Dulbecco's Modified Eagle Medium (DMEM) supplemented with 10% fetal bovine serum (FBS) and antibiotics, as mentioned earlier, for 15–20 days (*Suvà et al., 2014*).

## Reprogramming of DGCs
The differentiated counterparts of GSCs and differentiated GBM cell lines were removed from 10% FBS containing DMEM, spun down twice in phosphate-buffered saline (PBS) to remove any trace of FBS, and plated on ultra-low attachment plates in Neurobasal medium containing all the supplements mentioned earlier and antibiotics for 7–10 days.

## Neurosphere assay
Viral infection was done in the GSCs using lentivirus transduction for nontargeting shRNA (shNT) or shRNA for the gene of interest. The small pellets of cells were collected 48 hr after viral infection, dissociated to form single cells, counted, and replated in six-well plates ($3 \times 10^4$ cells/well) in complete Neurobasal medium. At the same time, cells were harvested and checked for specific gene manipulation (like knockdown verification). Media were replenished every 2–3 days, and neurosphere formation was monitored till the sixth or seventh day after replating. The number of neurospheres, sphere diameter, and size was analyzed using ImageJ software. Spheres having an area less than 50 µm$^2$ were excluded from the analysis.

## Limiting dilution assay
For each condition, 1, 10, 50, 100, and 200 GSCs (single cells) were plated in 10 wells each, respectively, of a 96-well plate, and sphere formation was assessed over the next 5–7 days. The number of wells not forming spheres was counted and plotted against the number of cells per well. Extreme limiting dilution assay was done using the ELDA software available online (https://www.elda.at/cdscontent/?contentid=10007.854970&portal=eldaportal).

## CM collection, concentration, and sample preparation for mass spectrometry
GSC cell lines were grown as neurospheres in a complete Neural Stem Cell medium containing glutamine, heparin, N2, B27, EGF, and FGF for 6 days. They were thoroughly washed using PBS and replated without disruption in Neural Stem Cell medium devoid of supplements and growth factors.

36 hr after plating in incomplete medium, the CM was collected, spun at 1500 rpm for 15 min, filtered using 0.45 µm syringe filters, and stored at –80°C. The DGCs were grown in complete DMEM supplemented with 10% FBS, then washed with PBS, and CM was collected after 36 hr in an incomplete medium and similarly stored.

The CM were concentrated using Centricons (3 kDa cutoff; Merck) from 8 ml to 100 µl, followed by precipitation of proteins with trichloroacetic acid (TCA 10% for 30 min at 4°C). Equal amounts of proteins from each condition were run on gradient (4–20%) SDS gels and stained with Colloidal Coomassie blue. The gels were then destained, and each lane was cut into five equal pieces. Proteins were digested in-gel using trypsin (Gold Promega, 1 µg per band, overnight at 30°C) as previously described (*Thouvenot et al., 2008*).

## Mass spectrometry, protein identification, and relative quantification

Tryptic peptides were analyzed online by nano-flow liquid chromatography coupled to tandem-mass spectrometry using a Q-Exactive+ mass spectrometer (Thermo Fisher Scientific, Waltham, USA) coupled to an RSLC-U3000 HPLC (Thermo Fisher Scientific). Desalting and pre-concentration of samples were performed online on a PepMap pre-column (0.3 mm × 10 mm, Dionex). A gradient consisting of 2–25% B for 80 min, 25–40% B for 20 min, and finally 40–90% B for 5 min (A = 0.1% formic acid; B = 0.1% formic acid in 80% acetonitrile) at 300 nl/min was used to elute peptides from the capillary reverse-phase column (0.075 mm × 150 mm, Acclaim PepMap 100 C18, Thermo Fisher Scientific). Eluted peptides were electro-sprayed online at a voltage of 1.5 kV into the Q-Exactive+ mass spectrometer. A cycle of one full-scan mass spectrum (MS1, 375–1500 m/z) at a resolution of 70,000, followed by 12 data-dependent tandem-mass (MS2) spectra, was repeated continuously throughout the nano-LC separation. Parameters used for MS2 spectra were the resolution of 17,500, AGC target of 1e5, normalized collision energy of 28, and an isolation window of 1.2 m/z. Mass spectra were processed using the MaxQuant software package (v 1.5.5.1, *Cox and Mann, 2008*) against the UniProtKB Reference proteome UP000005640 database for *Homo sapiens* (release 2018_11) and contaminant database. The following parameters were used: enzyme specificity set as trypsin/P with a maximum of two missed cleavages, oxidation (M) and acetyl (protein N-term) set as variable modifications, and carbamidomethyl (C) as fixed modification and a mass tolerance of 0.5 Da for fragment ions. The maximum false peptide and protein discovery rate was specified as 0.01. Match between runs was used to transfer identification from run to run. Relative protein quantification was performed using LFQ intensity in MaxQuant. The mass spectrometry proteomics data have been deposited to the ProteomeXchange Consortium via the PRIDE (*Perez-Riverol et al., 2022*) partner repository with the dataset identifier PXD032958. For statistical analysis, missing values were defined using the imputation tool of the Perseus software (v. 1.6.1.1, *Tyanova et al., 2016*).

## Lentivirus preparation and transduction of cells

HEK-293T cells were seeded in 60 mm or 90 mm poly-L-llysine-coated cell culture dishes. The cells were transfected with shRNA plasmid and helper plasmids psPAX2 and pMD2.G using Opti-MEM (Invitrogen #22600-050) medium and Lipofectamine (https://www.thermofisher.com/order/catalog/product/12566014; Invitrogen) when the cells were 60–70% confluent. 6 hr after transfection, the Opti-MEM medium was replaced by a fresh DMEM supplemented with 10% FBS. 60 hr post-transfection, the supernatant from the transfected cells was collected in 15 ml Falcon tubes, centrifuged at 5000 rpm for 10 min, filtered through 0.45 µm, and stored at –80°C for future use.

For the virus used in endothelial cells, DMEM was removed after 24 hr of transfection and changed to complete M199, and the virus was collected after 60 hr, as mentioned previously. The same method was used for the virus for GSCs (DMEM was changed to NBM). The shRNA construct number TRCN0000152163 from Sigma human TRC shRNA library was used for knockdown studies of human FMOD. The shRNA construct number TRCN000094248 from Sigma mouse TRC shRNA library was used for knockdown studies of mouse FMOD. A pooled lentivirus using the construct numbers TRCN000024441, TRCN000024441, TRCN000024441, and TRCN000024420 was used for knockdown studies of JAG1 in the ST1 cells.

## Endothelial cell culture

ST1 cells were grown in Medium 199 (Sigma #M4530) supplemented with 20% FBS and endothelial cell growth factor (ECGS) (Sigma #E2759), heparin, glutamine (1×) and 1× antibiotic-antimycotic solution (Sigma, # A5955, stock concentration 100×).

## Transdifferentiation of DGCs

The MGG8-DGCs were grown in Medium 199, having the same composition as mentioned above, for 3 days and subjected to hypoxia (1% $O_2$) for 8 hr. The cells then formed transdifferentiated endothelial cells (TDECs), showed endothelial morphology (data not shown), were harvested for checking markers levels, and were also plated for the in vitro angiogenesis assay.

## NICD stable generation

For NICD stable cell line generation, the ST1 cells were transfected with NICD pCMV Neo/intracellular domain of human Notch1 (NIC-1) plasmid using Lipofectamine 200 (Invitrogen). They then were harvested 48 hr after transfection for Western blot and in vitro angiogenesis assay.

## RNA isolation, cDNA conversion, and real-time quantitative RT-PCR analysis

Total RNA from the cells was isolated from cells using the TRI reagent (#T9424 Sigma) according to the manufacturer's instructions. Next, the integrity of the RNA was checked by running it on 2% MOPS-formaldehyde gel. RNAs were quantified by NanoDrop. 2 µg of total RNA were used for cDNA conversion using the high-capacity cDNA reverse transcription kit (Applied Biosystems, USA) according to the manufacturer's protocol. The cDNAs were diluted in a ratio of 1:10 with nuclease-free water to make their final concentration 10 ng/µl. Subsequently, real-time quantitative PCR was done using the ABI Quant Studio 5 and 6 (Life Technologies, USA). The cDNA was used as a template, and DyNAmo Flash SYBR Green qPCR kit (#F-416L) was used. Gene-specific primer sets were used for the reaction (see table at the end) under the following conditions: 95°C for 15 min, 40 cycles of 95°C for 20 s, 60°C for 25 s, and 72°C for 30 s followed by the dissociation cycle for melt curve generation. Each sample was run either in duplicate or triplicate. Glyceraldehyde 3-phosphate dehydrogenase (GAPDH), ACTB (beta-actin), 18S rRNA, RPL35a (ribosomal protein L35a), and *ATP5G1* (ATP synthase, H+transporting, mitochondrial F0 complex, subunit C1 [subunit 9]) were used as reference genes for human gene expression analysis. For mouse gene expression analysis, cyclophilin was used as a housekeeping gene. ΔΔCT method was used to calculate gene expression, which was transformed to log2 ratio and then to absolute scale for plotting.

## Western blotting

For Western blotting analysis, cell pellets were lysed in RIPA lysis buffer (containing 1 mM sodium orthovanadate, 5 mM sodium fluoride, 1 mM phenylmethanesulfonyl fluoride, and 1× protease inhibitor cocktail, Sigma, USA), and proteins were isolated from the cells by spinning at 13,000 rpm for 30 min. The supernatant containing the proteins was collected. Protein concentrations were measured using Bradford's reagent, and a standard BSA curve was used to determine the protein concentrations. Equal amounts of proteins from all conditions were mixed with protein loading dye (1×), denatured at 95°C for 15 min, loaded in each well of an SDS polyacrylamide gel, and the gel was run for around 8 hr. To prepare SDS- polyacrylamide gel, resolving and stacking gels were prepared at a concentration of 10–12 concentrations. The gel was run at 70–100 V, and then the proteins were transferred onto a polyvinyl fluoride (PVDF) membrane using the semi-dry transfer method. After the transfer, the membrane was blocked using 5% skimmed milk in 1× Tris-buffered saline-Tween (TBST) for 1 hr. Subsequently, the membrane was washed in TBST for 30 min and probed initially with primary antibodies in 5% BSA-TBST for 14–16 hr at 4°C. Then, the membrane was washed in TBST for 30 min, and the secondary antibody, diluted in 5% skimmed milk in TBST, was added and incubated at room temperature for 2–3 hr. Finally, the blot was washed and developed using PerkinElmer ECL Plus lightning and Bio-Rad Clarity and Clarity Plus ECL chemiluminescent reagent using GE Image Quant machine.

## Chromatin immunoprecipitation (ChIP)

ChIP assay was conducted with chromatin isolated from ST1 cells treated with 10 µM TGF-β RI inhibitor (SB431542) and DMSO for 6 hr. Briefly, after cross-linking, the nuclei were prepared and

sonicated to generate chromatin fragments between 100 and 10,000 bp following the manufacturer's protocol using the Simple Chromatin Immunoprecipitation kit (CST; Cat# 9003). The sheared chromatin was collected by centrifugation (10,000 × $g$ for 10 min at 4°C), and a 10 µl aliquot was removed to serve as a positive input sample. Aliquots of 100-µl-sheared chromatin were incubated with 2 µg of the required antibody/antibodies followed by Protein G magnetic beads for the stipulated time. An equal amount of IgG and H3 antibodies was used as control. The eluted DNA was analyzed by quantitative PCR using the FMOD promoter-specific primer set ChIP-F/ChIP-R (in the list of primers) to amplify the desired region in the FMOD promoter. Conditions of linear amplification were determined empirically for this primer. The PCR conditions were as follows: 95°C for 5 min; 95°C for 30 s, 56°C for 30 s, and 72°C for 30 s for 35 cycles. PCR products were resolved by electrophoresis on a 2% agarose gel and visualized after ethidium bromide staining. Real-time qPCR was performed with the same eluted DNA. The conditions were as follows: 95°C for 3 min; 95°C for 10 s, 56°C for 30 s, 72°C for 30 s for 40 cycles and 72°C for 5 min. The Ct values of different conditions were normalized to Ct values in IgG control.

## Boyden chamber assay for cell migration

Trans-well assay was done in 24-well Boyden chambers with 8 µm pore size polycarbonate membranes (BD Biosciences, San Diego, USA). ST1 cells (5 × 10⁴) were resuspended in 500 µl serum-free Medium 199 and placed in the upper chamber. The lower chamber was filled with serum-free Medium 199 with 750 µl CM or 400 nM final concentration of recombinant protein dissolved in an incomplete medium (serving as a chemoattractant). After 24 hr of incubation, cells remaining on the upper surface of the membrane were removed with a wet cotton bud. The cells that migrated to the lower surface of the membrane were fixed in ice-cold methanol, stained with crystal violet, and imaged in a light microscope.

## Boyden chamber assay for cell invasion

Trans-well assay was done in 24-well Boyden chambers with 8 µm pore size polycarbonate membranes coated with Matrigel (BD Biosciences). ST1 cells (5 × 10⁴) were resuspended in 500 µl serum-free Medium 199 and placed in the upper chamber. The lower chamber was filled with serum-free Medium 199 with 750 µl CM or 400 nM final concentration of recombinant protein dissolved in an incomplete medium (serving as a chemoattractant). After 24 hr of incubation, cells remaining on the upper surface of the membrane were removed with a wet cotton bud. The cells that have invaded the lower surface of the membrane were fixed in ice-cold methanol, stained with crystal violet, and imaged in a light microscope.

## Cell proliferation assay (MTT assay)

1.5 × 10³ ST1 cells were plated with 2% FBS-containing M199 Medium in each well of a 96-well plate. MTT assay was performed as per the established protocol. MTT was added to each well, and Formazan crystals formed after 3 hr of incubation were dissolved in DMSO, and the absorbance was measured at 420 nm. The first reading served as the untreated condition (0th time point). After this reading, the cells were treated with 400 nm rhFMOD or an equivalent concentration of bovine serum albumin (BSA), and readings were taken every 24 hr, till 96 hr. The cell viability was then plotted as a line graph.

## In vitro angiogenesis assay

ST1 endothelial cells were seeded in 96-well plates (10 × 10³–15 × 10³ cells per well), coated with Geltrex (Invitrogen), and grown in Medium 199 (Sigma) without growth factors. Equal protein amounts (50–100 µg) from serum-free CM from different conditions were added on top of the cells. After 10–12 hr of incubation, endothelial cells form tube-like structures. Each complete circular structure was considered one complete network (well-connected and not broken at any place). The total number of networks for each condition was counted double-blindly. For positive control, cells were plated in complete endothelial cell media (Medium 199) supplemented with ECGS and 20% FBS, and in the negative control, cells were plated in incomplete Medium 199 (without serum and ECGS).

## Immunofluorescence staining of fixed cells

Cells were plated on coverslips in 12-well plates and allowed to attach. The cells were then fixed with 4% paraformaldehyde (PFA) and permeabilized using PBS supplemented with 0.25% Triton-X100. Cells were then washed with PBS and blocked using PBS supplemented with 1% BSA, 0.3% Triton X-100, and 5% goat serum for 2 hr at room temperature. After blocking, the primary antibody, diluted in the blocking buffer, was added to the coverslips overnight at 4°C. For dual staining, the primary antibodies (one anti-mouse and the other anti-rabbit) were simultaneously added to the samples in the required dilutions. After removing the primary antibody, cells were thoroughly washed with PBS three times for 5 min each. Fluorescence-conjugated secondary antibody was dissolved in the blocking buffer and added to the cells for 3 hr, after which the cells were washed three times with PBS for 5 min. The cells were stained with DAPI (1 μl/ml) for 5 min at room temperature. Coverslips were mounted on glass slides using glycerol as a mounting agent and imaged using a Zeiss LSM 880 confocal microscope.

## Luciferase reporter assay

Cells were seeded in six-well plates and co-transfected with the reporter luciferase construct and pCMV-beta gal (as control) using Lipofectamine according to the manufacturer's instructions. 24–48 hr after transfection, cell extracts were made in reporter lysis buffer (Promega). Protein concentrations of the cell lysates were measured by Bradford assay reagent (Bio-Rad). 10 μg of protein were mixed with 30 μl of luciferase assay reagent (LAR) to determine the luciferase activity, and values were normalized to β-galactosidase activity units.

## Flow cytometry

The GSCs and DGCs are pelleted down and stained with live dead fluorescent stain (L34955 LIVE/ DEAD Fixable Violet Dead Cell Stain Kit, for 405 nm excitation) for 15 min at 37°C. Then, the cells are blocked using 10% FCS and 1% sodium azide in PBS. After blocking, the cells are washed with PBS thrice, incubated with the primary antibody for 2 hr at room temperature, washed again thrice, and incubated with the secondary antibody for 2 hr in the dark. The final pellet was washed thrice with PBS, dissolved in 200 μl 3% BSA in PBS, and analyzed using the BD FACS Verse flow cytometer. The data were analyzed using the FlowJo software.

## Reverse phase protein array analysis for the identification of differential protein expression upon FMOD treatment to endothelial cells

Untreated ST1 cells and cells treated for 10, 30, and 60 min with 400 nM rhFMOD were lysed using RPPA lysis buffer. Lysates were serially diluted in 5 twofold dilutions using lysis buffer and printed on nitrocellulose-coated slides using an Aushon Biosystem 2470 arrayer. Slides were probed with 304 validated primary antibodies followed by detection with appropriate biotinylated secondary antibodies. Slides were scanned, analyzed, and quantified using Array-pro Analyzer software (Media Cybernetics) to generate spot intensity (level 1 data). Signals were visualized by a secondary streptavidin-conjugated HRP antibody and DAB colorimetric reaction. The list of 304 antibodies can be found at http://www.mdanderson.org/research/research-resources/core-facilities/functional-proteomics-rppa-core/antibody-information-and-protocol.html.

## Cryo-sectioning of fixed mouse brain

Mice were perfused intracardiacally using 4% PFA solution, and the brains were harvested and stored in PFA for 12 hr and subsequently in 30% sucrose solution. The brains were then embedded in Poly-freeze solution (Sigma, #35059990) and sectioned into 20-μm-thick sections using a Leica Cryostat. The sections were stored at –80°C in Tissue Cutting Solution (TCS).

## Confocal microscopy-based immunofluorescence analysis of free-floating sections

The brain sections were removed from TCS, put in 96-well plates, and washed thoroughly with PBS for immunofluorescence. Then, the same protocol was followed for immunofluorescence staining of

tissue sections as that followed for monolayer cells grown on coverslips. At the final step, after adding the secondary antibody, DAPI or H33342, and PBS washes, the sections were individually mounted on glass slides using the ProLong Glass Antifade Mountant (Invitrogen, #P36980) and covered by coverslips. Images were taken using the Zeiss LSM 880 confocal microscope using ×10, ×20, and ×40 objectives for various conditions.

For immunofluorescence of FFPE sections, an extra step of antigen retrieval was performed by de-paraffinizing the sections in xylene, then boiling in distilled water twice for 5 min each. After this, the rest of the steps from permeabilization to mounting the same steps were followed as that of immunofluorescence of free-floating sections.

## Scoring methods for confocal images

The areas of the blood vessels and the fluorescence intensities for all the fluorophores used in the tissue sections were measured using the Zeiss Black software (https://www.zeiss.com/microscopy/int/products/microscope-software.html), converted to percentages, and plotted as bar diagrams. For determining the extent of vascular mimicry in the blood vessels of the tissues, the overlap of green (coming from GFP or GFAP staining of the tumor cells) and red (a measure of CD31 expression of the endothelial cells) forming yellow color was calculated as a measure of colocalization coefficients in the Zeiss Black software. The absolute values of the colocalization coefficients were plotted for each condition.

## Subcutaneous injection of DBT-Luc cells for the co-implantation mouse model

All animal procedures followed were approved by the Institute Ethical Committee for Animal Experimentation (Institute Animal Ethics Committee [IAEC] Project Number: CAF/Ethics/752/2020). Mouse cell line DBT-Luc-DGC, which grows as a monolayer cell line (stable for luciferase expression), was reprogrammed to form DBT-Luc-GSCs. A combination of $10^5$ DBT-Luc-GSCs (without any *shRNA*) and $10^6$ DBT-Luc-DGCs that carried either the *miRNT* or *miRFMOD* constructs (referred to as DBT-Luc-DGC/*miRNT* or DBT-Luc-DGC/*miRFMOD*, respectively) was injected subcutaneously in the mice. Two groups of mice (n = 5 each) received a combination of $10^5$ DBT-Luc-GSCs+$10^6$ DBT-Luc-DGC/*miRNT* and another two groups (n = 5 each) received a combination of $10^5$ DBT-Luc/GSCs +$10^6$ DBT-Luc-DGC/*miRFMOD* (only one of the two groups for both *miRNT* and *miRFMOD* received doxycycline). Doxycycline injection (intraperitoneal, 100 µg per animal) began on 9th day post the injection and was given on every day up to the 16$^{th}$ day, after which it was given on every alternate day till the end of the experiment. Since the doxycycline administration induced the *mCherry* expression along with the *shRNA* expression, the animals were imaged for both bioluminescence (marked by red on the timeline) and fluorescence (marked by orange on the timeline) on the days indicated in the timeline. Two other groups (n = 5 each) received either only $10^5$ DBT-Luc-GSCs or $10^6$ DBT-Luc-DGCs (not stables for any *shRNA*) as controls.

## Intracranial injection of GBM cells

Cells were harvested in incomplete DMEM or NBM depending on the cell type to be injected. $0.25 \times 10^6$ DGCs or $1 \times 10^5$ GSCs were injected intracranially (in the hippocampus, 2.5 mm deep) in each animal using a stereotaxic apparatus. The animals were imaged on the third day after injection and subsequently, every 5–6 days until the end of the experiment. In experiments using inducible shRNAs, doxycycline was injected every day for 10 days (from the 13th day after injection) and on every alternate day until the end of the experiment. MGG8-GSC (MGG8-GSC/shNT vs. MGG8-GSC/shFMOD), AGR53-GSCs (AGR53-GSC/*miRNT* vs. AGR53-GSC/*miRFMOD*), and DBT-Luc-GSCs (DBT-Luc-GSC/*miRFMOD*) cells were intracranially injected in this study.

## In vivo imaging

In vivo imaging was done for bioluminescence or fluorescence with the PerkinElmer IVIS Spectrum by using mild gas anesthesia (using isoflurane) for the animals.

## Hematoxylin and eosin staining

Brains from perfused mice were paraffin-embedded and sectioned using a microtome (Leica Biosystems, 5 µm sections). Sections were mounted on glass slides and removed from paraffin, rehydrated,

and stained with Harris Hematoxylin (Merck, 6159380051046) for nuclear staining and Eosin Y (SDFCL, 44027G25) solution for cytoplasmic staining. The sections were then mounted using DPX mounting medium and imaged at 0.8× using a Lawrence and Mayo digital microscope.

### Gene set enrichment analysis (GSEA)

The differentially expressed genes between the GSC and DGC (as identified in GSE54792) were preranked based on fold change and used as an input to perform GSEA. All the gene sets available in the Molecular Signature Database (MSigDB, roughly 18,000 gene sets) were used to run the GSEA. We filtered out the TGF-β pathway-related gene sets to identify that most of them were significantly enriched in the DGCs over the GSCs. Similarly, the same analysis was carried out in multiple publicly available GBM vs. normal samples datasets to show the significant enrichment of TGF-β gene sets in GBM over normal. We acknowledge our use of the GSEA software and MSigDB (*Subramanian et al., 2005*; http://www.broad.mit.edu/gsea/).

### Single sample GSEA (ssGSEA)

Gene set variation analysis (GSVA) was performed using ssGSEA to determine the enrichment of the different molecular subtypes of GBM in GSE54792 and also the enrichment of the TGF-β Hallmark gene set from MSigDb. The higher GSVA score indicates the highest enrichment, which gradually decreases.

### Survival analysis

Kaplan–Meier survival analysis was done using GraphPad Prism 5.0 (GraphPad Software, San Diego, CA).

### Heatmap generation

The heatmaps were generated using the Multiple Experiment Viewer (MEV) software (http://www.tm4.org/mev.html) version 4.8.1. LFQ values for protein expression from mass-spec data and mean pixel density for the RPPA were used as inputs for heatmap generation. A nonparametric *t*-test was performed with a false discovery rate (FDR) and a p-value cutoff of 0.05.

### Quantification and statistical analysis

Bar diagrams were generated using Microsoft Excel. The box plots were generated using GraphPad Prism 5.0 (GraphPad Software). The p-value was calculated by unpaired *t*-test with Welch's correction or Student's *t*-test done using Microsoft Excel indicated. ANOVA p-value was calculated using GraphPad Prism 5 software. Significance for Kaplan–Meier survival analysis as calculated by the log-rank test. Correlation coefficients were calculated using Pearson's correlation coefficient using GraphPad Software. All experiments (except the ones involving the animal models) were done at least thrice with duplicates for each condition within the experiments. The data are represented as mean ± standard deviation. p-Value <0.05 is considered significant with *, **, ***, and **** representing p-values <0.05, 0.01, 0.001, and 0.00001, respectively. ns indicates nonsignificance.

### List of primers used in this study

| 1 | FMOD | ACCTGCAGCTTGGAGAAGT | CAACACCAACCTGGAGAACC |
|---|---|---|---|
| 2 | SOX2 | AACCCCAAGATGCACAACTC | GCTTAGCCTCGTCGATGAAC |
| 3 | SALL2 | TAATCTCGGACTGCGAAGGT | TAGAACATGCGTTCTGGTGG |
| 4 | POU3F2 | TGACGATCTCCACGCAGTAG | GGCAGAAAGCTGTCCAAGTC |
| 5 | OLIG2 | CCAGAGCCCGATGACCTTTT | AGGACGACTTGAAGCCACTG |
| 6 | DLL4 | CTGCGAGAAGAAAGTGGACAGG | ACAGTCGCTGACGTGGAGTTCA |
| 7 | DLL3 | CACTCAACAACCTAAGGACGCAG | GAGCGTAGATGGAAGGAGCAGA |
| 8 | JAG1 | TGCTACAACCGTGCCAGTGACT | TCAGGTGTGTCGTTGGAAGCCA |

*Continued on next page*

*Continued*

| | | | |
|---|---|---|---|
| 9 | JAG2 | GCTGCTACGACCTGGTCAATGA | AGGTGTAGGCATCGCACTGGAA |
| 10 | HES1 | GGAAATGACAGTGAAGCACCTCC | GAAGCGGGTCACCTCGTTCATG |
| 11 | Mouse FMOD | CCCTTACCCCTATGAGCCC | GACAGTCGCATTCTTGGGGA |
| 12 | OSMR | CATCCCGAAGCGAAGTCTTGG | GGCTGGGACAGTCCATTCTAAA |
| 13 | ACSBG1 | GAACATCTGGTGCACGGTATAG | GAGGAAGCTGGTGGAGTATTG |
| 14 | ALDH1L | GAGGAAGCTGGTGGAGTATTG | ACGGTTGGCTGAAAGAAGAA |
| 15 | S100B | CCCTGTAGAAGAGTCACCTGTA | GCTGTGGGTCTGTAGATGTATG |
| 16 | GFAP | CGGAGACGCATCACCTCTG | AGGGAGTGGAGGAGTCATTCG |
| 17 | MELK | TATGAAACGATTGGGACAGGTG | CCCTAGCGCATTCTTATCCATGA |
| 18 | NESTIN | GGAATCTCTGAGGTCTCTTGATG | TCTGCTCCTCCTCTTCTACTT |
| 19 | BMI1 | CAAGAAGAGGTGGAGGGAATAC | CCAGAGAGATGGACTGACAAAT |
| 20 | KLF4 | GTGCCCCGACTAACCGTTG | GTCGTTGAACTCCTCGGTCT |
| 21 | TWIST2 | CCAAGGCTCTCAGAACAAGAA | GGAGACGTAAAGAACAGGAGTATG |
| 22 | PTGS2 | TGAGCAACTATTCCAAACCAGC | GCACGTAGTCTTCGATCACTATC |
| 23 | S100A6 | ATGGCATGCCCTCTGGATCAG | TTATTTCAGAGCTTCATTGTAGATC |
| 24 | Cyclophilin | CAGACGCCACTGTCGCTTT | TGTCTTTGGAACTTTGTCTG |
| 25 | ATP5G | CCAGACGGGAGTTCCAGAC | GACGGGTTCCTGGCATAGC |
| 26 | GAPDH | TTGTCAAGCTCATTTCCTGG | TGATGGTACATGACAAGGTGC |
| 27 | cPPT (for sequencing) | GAAGGAATAGAAGAAGAAGGT GGAGAG | |
| 28 | KLF8 | CCTGAAAGCTCACCGCAGAATC | TGCTTGCGGAAATGGCGAGTGA |
| 29 | FOXP1 | CAAAGAACGCCTGCAAGCCATG | GGAGTATGAGGTAAGCTCTGTGG |
| 30 | ETS1 | GAGTCAACCCAGCCTATCCAGA | GAGCGTCTGATAGGACTCTGTG |
| 31 | GATA4 | GCGGTGCTTCCAGCAACTCCA | GACATCGCACTGACTGAGAACG |
| 32 | ATF2 | GGTAGCGGATTGGTTAGGACTC | TGCTCTTCTCCGACGACCACTT |
| 33 | ITGB1 | GGATTCTCCAGAAGGTGGTTTCG | TGCCACCAAGTTTCCCATCTCC |
| 34 | ITGA6 | CGAAACCAAGGTTCTGAGCCCA | CTTGGATCTCCACTGAGGCAGT |
| 35 | ITGAV | AGGAGAAGGTGCCTACGAAGCT | GCACAGGAAAGTCTTGCTAAGGC |

## List of shRNAs used in the study (all IDs are of Sigma TRC whole-genome shRNA library)

| S. no | **Gene name** | Clone 1 | Clone 2 | Clone 3 | Clone 4 | Clone 5 |
|---|---|---|---|---|---|---|
| 1 | FMOD | TRCN0000153650 | TRCN0000156734 | TRCN0000152163 | TRCN0000153199 | TRCN0000151908 |
| 2 | Mouse FMOD | TRCN0000094246 | TRCN0000094245 | TRCN0000094248 | | |
| 3 | KLF8 | TRCN0000015878 | TRCN0000015879 | TRCN0000015880 | TRCN0000015881 | TRCN0000015882 |
| 4 | JAG1 | TRCN0000033439 | TRCN0000033440 | TRCN0000033441 | TRCN0000033442 | TRCN0000033443 |
| 5 | ITGA6 | TRCN0000057773 | TRCN0000057774 | TRCN0000057775 | TRCN0000057776 | TRCN0000057777 |
| 6 | ITGB1 | TRCN0000275134 | TRCN0000275133 | TRCN0000275083 | TRCN0000275135 | TRCN0000275082 |
| 7 | ITGAV | TRCN0000003238 | TRCN0000003239 | TRCN0000003240 | TRCN0000003241 | |

## List of inhibitors used

| S. no. | Inhibitor name | Target molecule | Catalog number |
|---|---|---|---|
| 1 | Gamma-secretase inhibitor (GSI) | Gamma secretase | 565750 (Merck) |
| 2 | RGD Peptide | Integrins | A5082 (Sigma-Aldrich) |
| 3 | PP2 | Src | P0042 (Sigma-Aldrich) |
| 4 | PP3 | Structural analog to PP2 | 529574 (Calbiochem) |
| 5 | H1152 | ROCK1 | 555550 (Calbiochem) |
| 6 | RAC1 inhibitor | RAC1 | 553502 (Sigma-Aldrich) |
| 7 | PF573228 | FAK | PZ0117 (Sigma-Aldrich) |
| 8 | SB431542 | TGFB RI | 616464 (Sigma-Aldrich) |

## Acknowledgements

The results published here are in whole or part, based upon data generated by The Cancer Genome Atlas pilot project established by the NCI and NHGRI. Information about TCGA and the investigators and institutions that constitute the TCGA research network can be found at http://cancergenome.nih.gov/. We acknowledge the shRNA consortium (Dr. Subba Rao), IISc, India, for shRNA constructs. The Central Animal Facility (CAF), Indian Institute of Science is acknowledged for all animal experiments. SS acknowledges IISc for fellowship. KS acknowledges CEFIPRA DBT, DST, ICMR, and CSIR (Govt. of India) for research grants. Infrastructure supported by DST FIST, DBT-IISc partnership program, and UGC is acknowledged. KS is awarded J. C. Bose Fellowship from DST. This research was supported by grants to DFM from the Israel Science Foundation (1315/15 and 1429/20). PM was supported by grants from Fondation pour la Recherche Médicale and CEFIPRA (IFC/5603-C/2016/503). Mass spectrometry experiments were carried out using the facilities of the Montpellier Proteomics Platform (PPM, BioCampus Montpellier).

## Additional information

### Funding

| Funder | Grant reference number | Author |
|---|---|---|
| Department of Biotechnology, Ministry of Science and Technology, India | | Kumaravel Somasundaram |
| Department of Science and Technology, Ministry of Science and Technology, India | | Kumaravel Somasundaram |
| Indo-French Centre for the Promotion of Advanced Research | n° IFC/5603-C/2016/503 | Kumaravel Somasundaram Philippe Marin |
| Israel Science Foundation | Grant no.1315/15 and 1429/20 | Dinorah Friedmann-Morvinski |
| Fondation pour la Recherche Médicale | | Philippe Marin |

The funders had no role in study design, data collection and interpretation, or the decision to submit the work for publication.

### Author contributions

Shreoshi Sengupta, Conceptualization, Data curation, Formal analysis, Investigation, Methodology, Resources, Validation, Visualization, Writing - review and editing; Mainak Mondal, Data curation,

Formal analysis, Methodology; Kaval Reddy Prasasvi, Animal Experiments, Formal analysis, Methodology, Validation, Writing - review and editing; Arani Mukherjee, Formal analysis, Methodology, Validation, Writing - review and editing; Prerna Magod, Animal Experiments, Methodology; Serge Urbach, Data curation, Formal analysis, Investigation, Methodology, Writing - review and editing; Dinorah Friedmann-Morvinski, Conceptualization, Methodology, Resources, Supervision, Writing - review and editing; Philippe Marin, Conceptualization, Funding acquisition, Methodology, Resources, Supervision, Writing - review and editing; Kumaravel Somasundaram, Conceptualization, Data curation, Formal analysis, Funding acquisition, Investigation, Methodology, Project administration, Resources, Supervision, Writing - original draft, Writing - review and editing

### Author ORCIDs
Dinorah Friedmann-Morvinski (ID) http://orcid.org/0000-0002-6394-9876
Philippe Marin (ID) http://orcid.org/0000-0002-5977-7274
Kumaravel Somasundaram (ID) http://orcid.org/0000-0001-6228-9741

### Ethics
The Institute Ethical Committee for Animal Experimentation (Institute Animal Ethics Committee [IAEC] Project Number: CAF/Ethics/752/2020).

### Decision letter and Author response
Decision letter https://doi.org/10.7554/eLife.78972.sa1
Author response https://doi.org/10.7554/eLife.78972.sa2

## Additional files

### Supplementary files
• Supplementary file 1. List of proteins exhibiting significantly different abundance in glioma stem-like cell (GSC) and differentiated glioma cell (DGC) secretomes.

• Supplementary file 2. List of significantly depleted gene sets from gene set enrichment analysis in glioma stem-like cell (GSC) vs. differentiated glioma cell (DGC) RNA-Seq Data (GSE54792).

• Supplementary file 3. ANOVA values between the different groups of the in vivo co-implantation mouse model experiment.

• Supplementary file 4. List of primers, shRNA clones, and inhibitors used in the study. (A) List of primers used in this study. (B) List of shRNAs used in the study; all IDs are of Sigma TRC whole-genome shRNA library. (C) List of inhibitors.

• MDAR checklist

• Source data 1. Raw images for blots of figures and figure supplements.

### Data availability
Label-free mass spectrometry data between the GSC and DGC showing protein ratios in the GSC and DGC secretome and p values are shown in Supplementary File 1 for proteins exhibiting significant differences in abundance in both conditions. The mass spectrometry proteomics data have been deposited to the ProteomeXchange Consortium via the PRIDE (Perez-Riverol et al., 2022) partner repository with the dataset identifier PXD032958.

The following dataset was generated:

| Author(s) | Year | Dataset title | Dataset URL | Database and Identifier |
|---|---|---|---|---|
| Somasundaram K | 2022 | Functional genomics of glioblastoma: from epigenetics to proteomic investigation of tumor-initiating cell secretome | https://www.ebi.ac.uk/pride/archive/projects/PXD032958 | EBI PRIDE, PXD032958 |

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

## Appendix 1

### The role of FMOD on GSC, DGC growth, and their plasticity in vitro

Before studying the importance of DGC secreted FMOD on tumor growth, we decided to investigate its requirement for the GSC, DGC growth, and their plasticity to form the other type. We have earlier shown that FMOD is not required for the proliferation of established glioma cell lines (*Mondal et al., 2017*). We used two human glioma cell lines, MGG8 and U251, two murine glioma cell lines, AGR53 (*Angel et al., 2020*) and DBT-Luc (*Yuan et al., 2007*).

MGG8-GSCs transduced with a small hairpin RNA targeting FMOD (MGG8-GSC/shFMOD) grew as neurospheres with equal efficiency as measured by neurosphere formation and limiting dilution assays and also differentiated to form DGCs as efficiently as control MGG8-GSCs transduced with non-targeting shRNA (MGG8-GSC/shNT) (*Figure 2—figure supplement 1A–D*). As expected, the differentiation was accompanied by the downregulation of glioma reprogramming factors in both MGG8-GSC/shNT and MGG8-GSC/shFMOD cells (*Figure 2—figure supplement 1*). In concordance with human GSCs, we found a higher expression of *FMOD* in AGR53-DGCs than AGR53-GSCs (*Figure 2—figure supplement 2A–C*). To silence the expression of *FMOD* in AGR53-DGC, we used a doxycycline-inducible *FMOD shRNA* (*miRFMOD*) construct that contains an inducible *mCherry-shRNA* cassette downstream of the Tet-responsive element (*Angel et al., 2020*; *Figure 2A*). Efficient silencing of *FMOD* in doxycycline-treated AGR53-DGC/*miRFMOD* cells was observed compared to AGR53-DGC/*miRNT* cells (*Figure 2—figure supplement 2D E*). Next, we investigated the impact of *FMOD* silencing on AGR53-GSC growth and differentiation to DGCs in vitro. Like human GSCs, AGR53-GSC/*miRNT* and AGR53-GSC/*miRFMOD* grew as neurospheres with equal efficiency both in the absence and presence of doxycycline (*Figure 2—figure supplement 3A D*). Further, both AGR53-GSC/*miRNT* and AGR53-GSC/*miRFMOD* differentiated efficiently to grow as a monolayer (*Figure 2—figure supplement 3C*). The differentiation resulted in the significant upregulation of astrocytic markers (*Figure 2—figure supplement 3*). These results confirm that FMOD is overexpressed in both human and murine DGCs but is not required for GSC growth and their differentiation to DGCs.

To study the role of FMOD on reprogramming of DGCs to form GSCs, AGR53-DGC and DBT-Luc DGC were tested for their ability to reprogram. DBT-Luc is a luciferase-expressing glioblastoma-derived cell line, which grows as a differentiated monolayer (referred to here onward as DBT-Luc-DGC) in an FBS-containing medium (*Pang et al., 2019*). AGR53-DGC/*miRNT* and AGR53-DGC/*miRFMOD* (both in the absence and presence of doxycycline) cells grow as a monolayer efficiently and reprogrammed to form neurospheres (*Figure 2—figure supplement 4A*). The reprogramming resulted in the significant upregulation of stem cell markers (*Figure 2—figure supplement 4B*). Similarly, DBT-Luc-DGC could readily reprogram to form DBT-Luc neurospheres (DBT-Luc-GSC; data not shown) with concomitant upregulation of stem cell markers and downregulation of astrocyte markers (*Figure 2—figure supplement 5A B*). DBT-Luc-DGCs expressed higher levels of *FMOD* transcript and protein compared to DBT-Luc-GSCs (*Figure 2—figure supplement 5C D*, respectively). To silence the expression of *FMOD* in DBT-Luc-DGC, we used the doxycycline-inducible *shFMOD* (*miRFMOD*) construct as explained above (*Angel et al., 2020*; *Figure 2A*). The addition of doxycycline resulted in downregulation of *FMOD* transcript and protein in DBT-Luc-DGC/*miRFMOD* but not in DBT-Luc-DGC/*miRNT* cells (*Figure 2—figure supplement 5E F*). As expected, both DBT-Luc-DGC/*miRFMOD* and DBT-Luc-DGC/*miRNT* cells showed mCherry expression after doxycycline treatment (*Figure 2—figure supplement 5G*, ). Both DBT-Luc-DGC/*miRNT* and DBT-Luc-DGC/*miRFMOD* (both in the absence and presence of doxycycline) cells reprogrammed to form neurospheres with equal efficiency (data not shown). We next explored the impact of FMOD silencing on the ability of U251 cells, an established human glioma cell line, to form neurospheres through reprogramming. U251 cells, which grow as differentiated monolayer cells (referred to here as U251-DGC) in an FBS-containing medium, can reprogram and form neurospheres enriched in CD133 expression (*Tao et al., 2018*). Both U251-DGC/shNT and U251-DGC/shFMOD cells could reprogram with equal efficiency, as measured by neurosphere formation and limiting dilution assays (*Figure 2—figure supplement 6A–C*). The neurospheres formed through reprogramming showed an upregulation of glioma reprogramming factors in U251-GSC/shNT and U251-GSC/shFMOD cells (*Figure 2—figure supplement 6D*). Collectively, these observations indicate that FMOD is not required for GSC or DGC growth and differentiation or reprogramming processes.

## Investigating integrin heterodimeric subunits that are required in rhFMOD-treated endothelial cells

Integrins are a family of α/β heterodimeric transmembrane adhesion receptors. To identify the integrin α and β subunits that are involved in rhFMOD activation of integrin signaling in endothelial cells, we resorted to an unbiased approach where we analyzed the transcriptome data of microvessels isolated through laser capture microdissection (LCM) from human brain samples (*Song et al., 2020*). The transcript abundance of all integrins subunits α subunits (n = 19) and β subunits (n = 13) is shown in the following table. Three integrin subunits with maximum transcript abundance – ITGA6, ITGB1, and ITGAV – were chosen for testing. The ability of rhFMOD to induce pFAK levels in ST1 cells silenced for any of the above three integrins was investigated. The results show that all three integrins are essential for rhFMOD activation of integrin signaling in endothelial cells (*Figure 4—figure supplement 3*).

**High-abundant integrins in endothelial cells**[*]

| S. no. | Microvessel 1 FPKM (log2) | Microvessel 2 FPKM (log2) | Microvessel 3 FPKM (log2) | Average FPKM (log2) |
|---|---|---|---|---|
| ITGA6 | 191 | 68 | 249 | 169 |
| ITGB1 | 78 | 26 | 151 | 85 |
| ITGAV | 57 | 91 | 61 | 70 |
| ITGA7 | 23 | 28 | 76 | 42 |
| ITGA1 | 30 | 4 | 79 | 38 |
| ITGB8 | 19 | 41 | 15 | 25 |
| ITGB5 | 39 | 9 | 21 | 23 |
| ITGA5 | 13 | 11 | 38 | 20 |
| ITGA10 | 11 | 9 | 40 | 20 |
| ITGB1BP1 | 16 | 7 | 26 | 16 |
| ITGA3 | 1 | 15 | 12 | 9 |
| ITGA2 | 11 | 4 | 8 | 8 |
| ITGA9 | 0 | 18 | 4 | 7 |
| ITGB2 | 12 | 6 | 2 | 7 |
| ITGB4 | 0 | 6 | 13 | 7 |
| ITGB3BP | 2 | 0 | 16 | 6 |
| ITGAE | 5 | 1 | 6 | 4 |
| ITGB7 | 0 | 0 | 11 | 4 |
| ITGB3 | 6 | 0 | 3 | 3 |
| ITGA4 | 7 | 0 | 0 | 2 |
| ITGAL | 1 | 0 | 5 | 2 |
| ITGA8 | 0 | 1 | 4 | 2 |
| ITGAX | 4 | 0 | 0 | 1 |
| ITGAM | 2 | 0 | 0 | 1 |
| ITGA11 | 1 | 0 | 0 | 0 |
| ITGB1BP2 | 0 | 0 | 1 | 0 |
| ITGA2B | 0 | 0 | 0 | 0 |
| ITGBL1 | 0 | 0 | 0 | 0 |

*Continued on next page*

*Continued*

| High-abundant integrins in endothelial cells[*] | | | | |
|---|---|---|---|---|
| ITGA9-AS1 | 0 | 0 | 0 | 0 |
| ITGAD | 0 | 0 | 0 | 0 |
| ITGB2-AS1 | 0 | 0 | 0 | 0 |
| ITGB6 | 0 | 0 | 0 | 0 |

*Song et al., 2020.

## Bioinformatics analysis of FMOD, HES1, and JAG1 transcript levels in GBM: Classification, correlation, and prognosis

The transcript data of FMOD, HES1, and JAG1 in different data sets were used for deriving (1) transcriptional upregulation in GBM over the control brain, (2) survival prediction by Kaplan–Meier analysis, and (3) correlation between transcripts (see following table for more details). We used a total of 1885 samples that included GBMs (n = 1833) and control brain samples (n = 52). Wherever the data is not available (NA), the specific analysis is not carried out. A p-value <0.05 is considered significant, with ****<0.0001, ***<0.001, **<0.01, and *<0.05. The nonsignificant data is denoted as 'ns.' The data is shown in *Figure 4—figure supplement 6* and *Figure 4—figure supplement 7*.

Expression, correlation, and survival of fibromodulin (FMOD), HES1, JAG1 in glioblastoma (GBM)

| S. no | Datas et | Source | Control | Survival information | Control brain (n) | GBM samples (n) | FMOD: GBM vs. control | | HES1: GBM vs. control | | JAG1: GBM vs. control | | FMOD survival Kaplan–Meier | | FMOD to HES1 correlation | | FMOD to JAG1 correlation | |
|---|---|---|---|---|---|---|---|---|---|---|---|---|---|---|---|---|---|---|
| | | | | | | | log2 fold change | p-Value | log2 fold change | p-Value | log2 fold change | p-Value | Hazard ratio | Low vs. high p-value | R-value | p-Value | R-value | p-Value |
| 1 | TCGA Affymetrix | TCGA | Yes | Yes | 10 | 528 | 6.87 | <0.000 | 8.98 | <0.000 | 8.64 | <0.000 | 1.40 | 0.00 | 30 | <0.0001 | 0.08 | 0.08 |
| 2 | TCGA Agilent | TCGA | Yes | Yes | 10 | 489 | 1.42 | 0.00 | 0.98 | <0.000 | 2.04 | <0.000 | 1.34 | 0.00 | 0.36 | <0.0001 | 0.17 | 0 |
| 3 | Gravendeel | GSE16011 | Yes | Yes | 8 | 159 | 1.9 | 0.00 | 1.42 | 0.00 | 1.84 | <0.000 | 2.01 | <0.0001 | 0.35 | <0.0001 | 0.12 | 0.15 |
| 4 | TCGA RNA Seq | TCGA | Yes | Yes | 4 | 156 | 2.91 | 0.00 | 1.01 | 0.01 | 2.32 | <0.000 | 1.31 | 0.13 | 0.22 | 0.01 | 0.05 | 0.58 |
| 5 | CGGA RNA Seq | CGGA | Yes | Yes | 20 | 105 | 12.94 | 0.01 | 9.62 | 0.01 | NA | | 2.25 | 0.00 | 0.06 | 0.58 | NA | |
| 6 | Lee Y | GSE13041 | No | Yes | NA | 191 | NA | | | | | | 1.46 | 0.01 | 0.36 | <0.0001 | 0.29 | <0.0001 |
| 7 | Oh | GSE58401 | No | No | NA | 105 | | | | | | | | | 0.33 | 0.00 | 0.25 | 0.01 |
| 8 | Bao | GSE48865 | No | No | NA | 100 | | | | | | | | | 0.57 | <0.0001 | 0.25 | 0.01 |

For survival prediction based on the methylation status of FMOD promoter, the samples were divided into methylation high (above median β value) and low (below median β value) for two CpG IDs, cg03764585 and cg04704856, derived from TCGA and GSE48461 datasets. The data is presented in *Figure 4—figure supplement 7*.

## Appendix 2

### Appendix 2—key resources table

| Reagent type (species) or resource | Designation | Source or reference | Identifiers | Additional information |
|---|---|---|---|---|
| Antibody | Anti-FMOD (rabbit polyclonal) | Abgent | Cat# AP9243b; RRID:AB_10612142 | WB (1:2000) |
| Antibody | Anti-FAK (rabbit polyclonal) | Cell Signaling Technology | Cat#T9026; RRID: AB_477593 RRID:AB_2269034 | WB (1:1000) |
| Antibody | Anti-phospho FAK (Tyr397) (rabbit polyclonal) | Cell Signaling Technology | Cat# 3283; RRID:AB_2173659 | WB (1:500) |
| Antibody | Anti-GAPDH (mouse monoclonal) unconjugated, clone GAPDH-71.1 | Sigma-Aldrich | Cat# G8795; RRID:AB_1078991 | WB (1:20,000) |
| Antibody | Anti-actin (mouse beta monoclonal), horseradish peroxidase conjugated, clone AC-15 | Sigma-Aldrich | Cat# A3854; RRID:AB_262011 | WB (1:20,000) |
| Antibody | Anti-HES1 (D6P2U) (rabbit monoclonal) | Cell Signaling Technology | Cat# 11988; RRID:AB_2728766 | WB, IHC (1:1000) |
| Antibody | Anti-Von Willebrand factor (rabbit polyclonal) | Abcam | Cat# ab6994; RRID:AB_305689 | IHC (1:2000) |
| Antibody | Anti-CD31 (mouse monoclonal) | Cell Signaling Technology | Cat# 89C2; RRID:AB_2160882 | IHC (1:200) |
| Antibody | Anti-Jagged1 (rabbit monoclonal) | Cell Signaling Technology | Cat# 70109; RRID:AB_2799774 | IHC (1:200) WB (1:1000) |
| Antibody | Anti-FMOD (rabbit polyclonal) | Invitrogen | Cat# PA5-26250; RRID:AB_2543750 | IHC (1:100) WB (1:1000) |
| Antibody | Anti-CD133 (rabbit monoclonal) | Abcam | Cat# ab216323; RRID:AB_2847920 | Flow cytometry (1:100) |
| Antibody | Anti-Prom1 (mouse monoclonal) | Invitrogen | Cat# MA1-219; RRID:AB_2725113 | IHC (1:100) |
| Antibody | Anti-SOX2 (rabbit monoclonal) | Cell Signaling Technology | Cat# 3579; RRID:AB_2195767 | IHC (1:100) |
| Antibody | Anti-GFAP (mouse monoclonal) | Abcam | Cat# ab279290; RRID:AB_1209224 | ICC (1:200) |
| Antibody | Anti-GFAP (rabbit polyclonal) | Abcam | Cat# ab7260; RRID:AB_305808 | IHC (1:200) |
| Antibody | Anti-NOTCH1 (rabbit polyclonal) | Invitrogen | Cat# PA5-99448; RRID:AB_2818381 | WB (1:200) |
| Antibody | Anti-SMAD2 (rabbit monoclonal) | Cell Signaling Technology | Cat# 5339; RRID:AB_10626777 | WB (1:1000) |
| Antibody | Anti-pSMAD2 (rabbit monoclonal) | Cell Signaling Technology | Cat# 18338; RRID:AB_2798798 | WB (1:1000) |
| Antibody | Integrin beta-1 (rabbit monoclonal) | Cell Signaling Technology | Cat# 9699; RRID:AB_11178800 | WB (1:1000) |
| Antibody | Anti-integrin alpha-V (rabbit monoclonal) | Cell Signaling Technology | Cat# 4711; RRID:AB_2128178 | WB (1:1000) |
| Antibody | Anti-integrin alpha 6 (rabbbit monoclonal) | Abcam | Cat# ab18155 | WB (1:500) |
| Antibody | KLF8 antibody (rabbit polyclonal) | Abcam | Cat# ab168527 | WB (1:500) |

*Appendix 2 Continued on next page*

*Appendix 2 Continued*

| Reagent type (species) or resource | Designation | Source or reference | Identifiers | Additional information |
|---|---|---|---|---|
| Antibody | Goat anti-mouse HRP conjugate (mouse polyclonal) | Bio-Rad | Cat# 170-5047; RRID:AB_11125753 | WB (1:5000) Secondary antibody |
| Antibody | Goat anti-rabbit (H+L) HRP conjugate (rabbit polyclonal) | Invitrogen | Cat# 31460; RRID:AB_228341 | WB (1:5000) Secondary antibody |
| Antibody | Goat anti-mouse IgG (H+L), Alexa Fluor 488 (mouse polyclonal) | Invitrogen | Cat# A-11029; RRID:AB_138404 | IHC and ICC (1:500) Secondary antibody |
| Antibody | Goat anti-rabbit IgG (H+L), Alexa Fluor 488 (rabbit polyclonal) | Invitrogen | Cat# A-11034; RRID:AB_2576217 | IHC and ICC (1:500) Secondary antibody |
| Antibody | Goat anti-mouse IgG (H+L), Alexa Fluor 594 (mouse polyclonal) | Invitrogen | Cat# A-11032; RRID:AB_2534091 | IHC and ICC (1:500) Secondary antibody |
| Antibody | Goat anti-rabbit IgG (H+L), Alexa Fluor 594 (rabbit polyclonal) | Invitrogen | Cat# A-11037; RRID:AB_2534095 | IHC and ICC (1:500) Secondary antibody |
| Antibody | Goat anti-rabbit IgG (H+L), Alexa Fluor 405 Plus (rabbit polyclonal) | Invitrogen | Cat# A48254; RRID:AB_2890548 | IHC and ICC (1:500) Secondary antibody |
| Antibody | Goat anti-mouse IgG (H+L), Alexa Fluor 405 Plus (mouse polyclonal) | Invitrogen | Cat# A48255; RRID:AB_2890536 | IHC and ICC (1:500) Secondary antibody |
| Biological sample (*Mus musculus*) | Healthy adult C57BL/6 brain tissue | This paper | N/A | Female, whole brain, isolated from the mice after perfusion for sectioning; check 'Cryo-sectioning of fixed mouse brain' section |
| Biological sample (*M. musculus*) | GBM adult C57BL/6 brain tissue | This paper | N/A | Female, whole brain, isolated from the mice after perfusion for sectioning; check 'Cryo-sectioning of fixed mouse brain' section |
| Biological sample (*M. musculus*) | NU/J Healthy adult nude mice brain tissue | This paper | N/A | Female, whole brain, isolated from the mice after perfusion for sectioning; check 'Cryo-sectioning of fixed mouse brain' section |
| Biological sample (*M. musculus*) | NU/J GBM adult nude mice brain tissue | This paper | N/A | Female, whole brain, isolated from the mice after perfusion for sectioning; check 'Cryo-sectioning of fixed mouse brain' section |
| Biological sample (*M. musculus*) | GBM adult C57BL/6 subcutaneous tumor tissue | This paper | N/A | Female, whole brain, isolated from the mice after perfusion for sectioning; check 'Cryo-sectioning of fixed mouse brain' section |
| Chemical compound, drug | $\gamma$-Secretase inhibitor (GSI) | Merck | 565750 | N/A |
| Chemical compound, drug | RGD peptide (integrin inhibitor) | Sigma-Aldrich | A8052; CAS# 99896-85-2 | N/A |
| Chemical compound, drug | FAK inhibitor (PF-573228) | Sigma-Aldrich | PZ0117; CAS# 869288-64-2 | N/A |
| Chemical compound, drug | PP2 (Src inhibitor) | Sigma-Aldrich | P0042; CAS# 172889-27-9 | N/A |

*Appendix 2 Continued on next page*

*Appendix 2 Continued*

| Reagent type (species) or resource | Designation | Source or reference | Identifiers | Additional information |
|---|---|---|---|---|
| Chemical compound, drug | PP3 (negative control, structural analog of PP2) | Sigma-Aldrich | 529574; CAS# 5334-30-5 | N/A |
| Chemical compound, drug | ROCK1 inhibitor | Sigma-Aldrich | 555550; CAS# 871543-07-6 | N/A |
| Chemical compound, drug | RAC1 inhibitor | Sigma-Aldrich | 553502; CAS# 1177865-17-6 | N/A |
| Chemical compound, drug | TGF-β-R1 inhibitor (SB431542) | Sigma-Aldrich | 616464; CAS# 301836-41-9 | N/A |
| Peptide, recombinant protein | Fibromodulin (FMOD) (NM_002023) human | Origene | CAT# TP306534 | Recombinant protein |
| Peptide, recombinant protein | FMOD wild-type peptide (RLDGNEIKR) | Cellmano Biotech Limited, China | N/A | Peptide, custom made upon order |
| Peptide, recombinant protein | FMOD mutant peptide (RLDGNQIMR) | Cellmano Biotech Limited, China | N/A | Peptide, custom made upon order |
| Cell line (*Homo sapiens*) | LN229 | The American Type Culture Collection | #CRL-2611; RRID:CVCL_0393 | N/A |
| Cell line (*H. sapiens*) | U251-MG | European Collection of Authenticated Cell Cultures | #09063001; RRID:CVCL_0021 | N/A |
| Cell line (*H. sapiens*) | MGG4, MGG6. MGG8 | Kind gift from Dr. Wakimoto, Massachusetts General Hospital, Boston, USA | N/A | N/A |
| Cell line (*M. musculus*) | AGR-53 | Kind gift from Dr. Dinorah Friedmann-Morvinski, Tel Aviv University, Israel | N/A | N/A |
| Cell line (*M. musculus*) | DBT-Luc | Kind gift from Dr. Dinesh Thotala, Washington University in St. Louis, St Louis, MO, USA | N/A | N/A |
| Cell line (*H. sapiens*) | ST1 (human pulmonary microvascular endothelial cells [HPMECs]) | Kind gift from Dr. Ron Unger, Johannes Gutenberg University, Germany. | N/A | N/A |
| Cell line (*H. sapiens*) | Human brain-derived microvascular endothelial cells (HBMECs) | Cell Biologics, USA | #H-6023 | Primary Cell Line |
| Cell line (*M. musculus*) | B.End3 mouse brain-derived immortalized endothelial cells | The American Type Culture Collection | #CRL-2299; RRID:CVCL_0170 | N/A |
| Cell line (*H. sapiens*) | U87-MG | European Collection of Authenticated Cell Cultures | #89081402; RRID:CVCL_0022 | N/A |
| Genetic reagent (*M. musculus*) | C57BL/6J | The Jackson Laboratory | JAX:000664; RRID:IMSR_JAX:000664 | Female, 6–8 weeks old |
| Genetic reagent (*M. musculus*) | NU/J (athymic nude) | The Jackson Laboratory | JAX 002019; RRID:IMSR_JAX:002019 | Female, 6–8 weeks old |
| Recombinant DNA reagent | RAR3G (TetOn) | Dr. Dinorah Friedmann-Morvinski, Tel Aviv University | N/A | Plasmid |
| Recombinant DNA reagent | FMOD overexpression construct in the pcMV-entry backbone | Origene | #LY419579 | Plasmid |

*Appendix 2 Continued on next page*

*Appendix 2 Continued*

| Reagent type (species) or resource | Designation | Source or reference | Identifiers | Additional information |
|---|---|---|---|---|
| Recombinant DNA reagent | CSL Luc | Prof. Thomas Kandesch, Department of Genetics, University of Pennsylvania School of Medicine | N/A | Plasmid |
| Recombinant DNA reagent | pHes1(2.5k)-luc | Addgene | Cat# 43806 | Plasmid |
| Recombinant DNA reagent | NICD pCMV Neo/ intracellular domain of human Notch1 (NIC-1) | Prof. Thomas Kandesch, Department of Genetics, University of Pennsylvania School of Medicine | N/A | Plasmid |
| Recombinant DNA reagent | pMD2.G | Dr. G.S. Rao, Indian Institute of Science | N/A | Plasmid |
| Recombinant DNA reagent | psPAX2 | Dr. G.S. Rao, Indian Institute of Science | N/A | Plasmid |
| Recombinant DNA reagent | VSVG | Dr. Dinorah Friedmann-Morvinski, Tel Aviv University | N/A | Plasmid |
| Recombinant DNA reagent | pREV | Dr. Dinorah Friedmann-Morvinski, Tel Aviv University | N/A | Plasmid |
| Recombinant DNA reagent | pMDL | Dr. Dinorah Friedmann-Morvinski, Tel Aviv University | N/A | Plasmid |
| Software, algorithm | ImageJ | ImageJ (http://imagej.nih.gov/ij/) | RRID:SCR_003070 | N/A |
| Software, algorithm | MaxQuant software package | MaxQuant (https://www.maxquant.org/) | RRID:SCR_014485 | V 1.5.5.1 |
| Software, algorithm | Zen Black from Zeiss | Zen Black (https://www.zeiss.com/microscopy/int/products/microscope-software/zen.html) | RRID:SCR_018163 | N/A |
| Software, algorithm | GraphPad Prism | GraphPad Software (https://www.graphpad.com) | RRID:SCR_002798 | Version 6.01 for Windows |
| Software, algorithm | FlowJo | FlowJo (https://www.flowjo.com/solutions/flowjo) | RRID:SCR_008520 | 9.9.6 and v10 |
| Other | Harris Hematoxylin | Merck | 6159380051046 | Histological stain |
| Other | Eosin Y | SDFCL | 44027G25 | Histological stain |
| Other | Reverse phase protein array (RPPA) | https://www.mdanderson.org/research/research-resources/core-facilities/functional-proteomics-rppa-core.html | RRID:SCR_016649 | M.D. Anderson Cancer Center, University of Texas, USA |

