## [Editor Report]

The authors shed light on the role that differentiated glioma cells exerts in promoting cancer progression, revealing that the secreted fibromodulin by differentiated glioma cells is crucial in mediating angiogenesis in glioma via integrin-dependent Notch signaling. The results are important for gaining insight into the less concerned differentiated glioma cells in promoting cancer and would potentially enrich the treatment strategy for glioma.

---

## [Decision Letter]

**Decision letter after peer review:**

Thank you for submitting your article "Non-cancer stem cell-derived Fibromodulin activates Integrin-dependent Notch signaling in endothelial cells to promote tumor angiogenesis and growth" for consideration by *eLife*. Your article has been reviewed by 3 peer reviewers, including Caigang Liu as the Reviewing Editor and Reviewer #1, and the evaluation has been overseen by Mone Zaidi as the Senior Editor. The following individual involved in review of your submission has agreed to reveal their identity: Herve Chneiweiss (Reviewer #2).

Essential revisions:

1) Non-CSC as the key formulation needs to be specified and clearly clarified;

2) Statement regarding the biomarker of GSC in the Introduction part may require correction, and some important findings should be included as Reference;

3) Outline the reasons for the selection of the employed cell lines;

*Reviewer #2 (Recommendations for the authors):*

1) Be more precise on why you select a cell type and a given method. May be some results are not absolutely needed and removing them could ease reading the manuscript

2) Justify why the patient-derived cell lines were not mainly used along the work and why you used "circus" cell lines such as U87 or U251.

---

## [Author Response]

Essential revisions:1) Non-CSC as the key formulation needs to be specified and clearly clarified;

We understand the possibility that the use of "non-cancer-stem cells (non-CSCs)" may create confusion as it may sound like referring to stromal cells of cancer that are non-cancer cells of the tumor. This was rightly pointed out by reviewer # 2. After obtaining advice from Dr. Caigang Liu, reviewing Editor, we decided to use "glioma stem-like cells" (GSCs) and "differentiated glioma cells" (DGCs) in place of CSCs and non-CSCs respectively.

We also considered the Reference (Qiao X, et al. Association of human breast cancer CD44-/CD24- cells with delayed distant metastasis. e*Life*. 2021 Jul 28;10:e65418) referred by Dr. Caigang Liu, reviewing Editor for formulating the non-CSCs. This is a good suggestion.

However, we thought it would be difficult to use this terminology as many GSC markers for glioma have been identified, resulting in heterogeneity within the GSC population.

2) Statement regarding the biomarker of GSC in the Introduction part may require correction, and some important findings should be included as Reference;

The statement regarding biomarker of GSC is addressed appropriately, taking the recommendations of Reviewer #2. Accordingly, references that reported the stemness properties of CD133- glioma cells (Beier et al., 2007; Chen et al., 2010; Joo et al., 2008; Ogden et al., 2008; Wang et al., 2008), involvement of miR302-367 cluster and miR18A* in the regulation of stemness (Fareh et al., 2012, Turchi et al., 2013), and the existence of stem cell-associated heterogeneity in GBM (Dirkse et al., 2019) are referred in the revised manuscript.

3) Outline the reasons for the selection of the employed cell lines;

1) We have used three patient-derived GSC lines (MGG4, MGG6, and MGG8) for the initial discovery and in vitro validation of the DGC-specific expression of FMOD.

Please refer to Main Figure 1.

2) We have used one human patient-derived GSC line (MGG8) and two murine GSC lines (AGR53 and DBT-Luc) in an intracranial orthotopic mouse model experiment to prove the importance of FMOD induced angiogenesis in tumor growth. While MGG8 line silenced for FMOD using short hairpin RNA (shRNA) was used to prove the importance of FMOD in tumor angiogenesis and growth (Please refer to Main Figure 6), AGR53 and DBT-Luc lines carrying doxycycline-inducible shRNAs proved the importance of FMOD secreted by de novo generated DGCs from a GSC initiated tumor in angiogenesis and growth.

Please refer to Main Figure 5 and Main Figure 5—figure supplement 1, respectively.

3) We have also used the murine GSC line (DBT-Luc-GSC) for co-implantation experiment Please refer to Main Figure 2.

4) The only mouse model experiment carried out using the U251 cell line is now moved to the supplementary section as these cell lines are highly unrepresentative since they are grown for a long time in serum conditions, as recommended by reviewer # 2

Please refer to Main Figure 6—figure supplement 1.

5) The other places where the established glioma cell lines, LN229, U251, and U87 used, are in the main figure 3 and its figure supplements. In these experiments, we used these cells only as a source of secreted FMOD to study the angiogenesis-inducing property. However, we would like to point out that conditioned media from MGG4, MGG6, and MGG8 also showed the angiogenesis-inducing property of FMOD.

Please refer to Main Figure 3 and its figure supplements.

[Editors' note: we include below the reviews that the authors received from another journal, along with the authors’ responses.]

We thank the reviewers for their constructive comments, based on which we have revised the manuscript. All questions have been appropriately addressed and additional data have been included when requested. Please find below point-to-point responses to the reviewers’ comments describing how they were addressed.

Answers to comments from Reviewer 11) Large GBM or therapy-resistant GBM is characterized by at least three forms of neovascularization, angiogenesis, vasculogenesis, and vascular mimicry. This manuscript only dealt with angiogenesis. GSC is known to be involved in pericyte development to support angiogenesis and vasculogenesis. Recent reports also indicate the involvement of GSC in making vascular mimicry. The authors should at least show the effect of FMOD on vascular mimicry in vitro or in vivo (I believe the authors already have tissues from different conditions and staining for VM markers will prove the relationship).

In the first version, we showed the important role played by FMOD on tumor angiogenesis as measured by the ability of endothelial cells to form angiogenic networks. In the revised manuscript, we extended our studies to vascular mimicry, another form of neovascularization wherein the tumor cells transdifferentiate to form endothelial cells (TDECs) and obtain more nutrients.

– in vitro, we demonstrate that MGG8-DGCs and U87 readily differentiated to form TDECS when grown in M199 media, which correlated with increased CD31 expression and decrease in FMOD expression. Both TDECs formed angiogenic networks more efficiently in the presence of rhFMOD. *These data are illustrated on Figures 3G-J and Supplementary Figure 15. See page no. 9, lines 259 to 271 for more detail in the revised manuscript.*

– in vivo, we demonstrate that large tumors formed by MGG8-GSC, AGR53-GSC, and DBTLuc-GSC in FMOD+ condition show the presence of TDECs, as evidenced by significant colocalization of CD31 and either GFAP (for MGG8) or GFP (for AGR53 and DBT-Luc) on blood vessels. This process was significantly reduced in small tumors formed by the above cells in FMOD-silenced conditions. *These data are illustrated on Figures 7 G-L and Supplementary Figures 27G-I; see page no. 14, lines 422 to 428 for more detail in the revised manuscript.*

2) Larger GBM and recurrent GBM release many cytokines and attractants that mobilize and accumulate BMDPC in the tumor and cause vasculogenesis. The author should investigate or discuss the effect of FMOD on vasculogenesis.

We appreciate the suggestion of the reviewer to investigate the role of BMDPC and vasculogenesis. We tried to examine this phenomenon by staining the tumor sections using antiCD34 (a BMDPC-specific endothelial cell marker). While we found colocalization of CD34 and CD31 in a few locations on the blood vessels (the data are shown in Author response image 1), we are not comfortable with the quality of the CD34 antibody available. Hence, we would that these results are not included in the manuscript and to consider this line of investigation in the future.

**Author response image 1. sa2fig1:** Co-localization of BMDPC marker CD34 and endothelial marker CD31 in blood vessels. A. Co-localization of the two markers in tumor sections of MGG8-GSC/shNT. B. Co-localization of the two markers in MGG8-GSC/shFMOD tumor sections. Yellow regions in the merged image of MGG8-GSC/shNT tumor section depicts the co-localization of CD34 and CD31. Scale = 50 µm, Magnification = 40X.

3) ST1 is an endothelial cell line that originated from the lungs. These are not a replica of GBM or brain associated microvascular derived EC. The author should prove that there are no differences between EC from the brain and lungs in respect of in vitro angiogenesis with or without FMOD.

To address this issue, we have used two endothelial cell lines originating from human and murine brain -primary human brain microvascular endothelial cells (HBMECs) and murine immortalized brain endothelial cells (B.End3), respectively. Our results show that CM derived from MGG8 DGC/shNT is more efficient to promote formation of angiogenic networks by both B.End3 and HBMECs than CM derived from MGG8 DGC/shFMOD cells (Supplementary Figure 14F; Figures 3E and F). The effect of FMOD silencing was rescued by adding rhFMOD exogenously to both cell types (Supplementary Figure 14F; Figures 3E and F). This indicates that the pro-angiogenic effect of FMOD is conserved across endothelial cells, irrespective of their tissue of origin, and validates our initial findings using the human pulmonary microvascular endothelial cells (ST1). See page no. 9, lines no. 253 to 258 for more detail in the revised manuscript.

4) If the development of GBM from GSC is related to DGC, then targeting GSC will not be the answer for therapeutic purposes. The authors should give a guideline on how to deal with the scenario in the clinical context. Antiangiogenic therapy is not fruitful, therefore, the guideline should emphasize breaking the link between DGC and GSC.

The present study demonstrates that in addition to GSCs, DGCs have an essential role in tumor growth and maintenance. While the therapy-resistant and self-renewing GSCs trigger the early events of transformation and growth, DGCs, the proportion of which continues to increase during tumor growth, progressively become essential. Thus, targeting both cancer stem cells and differentiated cancer bulk cells is crucial to achieve a durable response. The study also highlights the potential of GSC and DGC CM analysis to uncover novel targets in cancer therapy and the critical influence of DGC-secreted FMOD in glioma tumor growth. This aspect is discussed in the Discussion section. See page no. 17, lines no. 504 to 510 for more detail in the revised manuscript.

Answer to comments from Reviewer 21) The manuscript is very hard to follow in places and takes leaps of logic. It would be far more helpful if the authors would focus on generating data to support a coherent story that is logically presented.

To address this issue, we have modified the manuscript to make it more coherent. The experiments showing that FMOD is not required for GSC growth and differentiation as well as DGC growth and reprogramming have been moved to supplementary information. These data are now illustrated on supplementary figures 6-11.

2) One issue is that the proteomics study seems a bit too easy that the authors happened to identify a protein that they have already studied. It would be better to do a secondary screen or some other prioritization.

The choice of FMOD from the proteomic data was motivated by the objective to identify proteins preferentially secreted by DGCs involved in ECM remodelling. This is the reason why we considered proteins enriched in the DGC secretome, compared to the GSC secretome, and annotated to be ECM proteoglycans (see the Venn diagram on Figure 1B). Among the six proteins exhibiting these characteristics, FMOD was the most enriched in the DGC secretome, compared to the GSC secretome. These results and our earlier work on FMOD led us to choose FMOD to further investigate its role in tumor formation by inducing ECM remodelling and associated phenotypes such as angiogenesis.

See page no. 5 and 6, lines no 148 to 165 for more detail in the revised manuscript.

3) One major concern is that GSC and DGC are cultured differently. It is not clear to me that FMOD is strictly connected to differentiated cells, but rather adherent cells grown in serum, which can induce ECM expression. The in vivo studies in Figure 2 are the best evidence but should be better developed.

Several lines of evidence support that “specific” FMOD expression in DGCs (compared with GSCs) is not due to specific culture conditions (grown as adherent cells in serum-containing medium):

1) Our previously published study (Mondal et al., 2017, Oncogene, 36, 71–83) showed a highly variable pattern of FMOD expression by several established adherent GBM cultures even though all cell lines used in the study were grown as adherent cultures in serum-containing medium. U251, U87, and U343 cells show very high FMOD expression while others like LN229 and LN18 cells show minimal or no detectable FMOD expression. This observation suggests that FMOD expression is not related to specific culture conditions such as the use of adherent cells and serum-containing media.

2) Both GSCs and DGCs (MGG8) are grown as a monolayer on Geltrex-coated cell culture dishes. In this condition, we show that (see Author response image 2 for detail):

a) FMOD RNA and protein (intracellular and secreted) are still more abundant in DGCs than in GSCs (see panels A and B of Author response image 2).

**Author response image 2. sa2fig2:** Figure 2: GSCs and DGCs grown as adherent cultures retain expression of their unique markers. A. Real-time qRT-PCR analysis showing FMOD mRNA upregulation in DGCs over GSCs. B. Western blotting showing FMOD protein overexpression, both in the total cellular proteome (left) and the secretome (right) of DGCs vs. GSCs. C. Real-time qRT-PCR analysis showing downregulated expression of four reprogramming factors in DGCs compared with GSCs. D. Expression of SSEA1, FMOD, CD133, and GFAP in GSCs vs. DGCs in adherent conditions. E. Flow cytometry analysis showing and enrichment of CD133-positive cells in the GSC cultures compared with DGC cultures.

b) RNA level of four reprogramming factors are upregulated in GSCs, compared with DGCs (see panel C of Author response image 2).

c)Confocal staining of SSEA1, *SOX2*, and CD133 show that they are upregulated in GSCs, while GFAP and FMOD are upregulated in DGCs (see panel D of Author response image 2).

d) FACS analysis of CD133 expression by GSCs and DGCs shows an enrichment of CD133 positive-cells in the GSC monolayer but not in the DGCs (see panel E of Author response image 2).

Our results show that even when GSCs and DGCs are grown in adherent conditions, DGCs express higher levels of FMOD than GSCs. Thus, the DGC-specific expression of FMOD compared to GSC is not related to adherent growth conditions.

3) We also demonstrate that in FMOD+ tumors (in all three orthotopic tumors using AGR53, DBT-Luc, and MGG8 GSCs), FMOD colocalizes with GFAP, further supporting that preferential FMOD expression and secretion by DGCs takes place in vivo*.*

These data are illustrated on Supplementary Figures 24, 25, and 26;. See page no. 14, lines no. 405 to 414 for more detail in the revised manuscript.

4) FMOD has been strongly connected to TGF-β signaling (and invasion and angiogenesis). It is not clear from the studies how the proposed signaling integrates into the prior modeling of protein function.

As suggested by the reviewer, we firmly believe that FMOD expression in DGCs is under the control of TGF-β signaling. In the revised manuscript, the following aspects have been addressed:

1) GSEA analysis of DEGs (differentially regulated genes) in GSCs over DGCs (Suva et al., 2014, 157, 580-594, Cell) showed a significant depletion of many gene sets associated to TGF-β signaling (Supplementary Figure 2).

2) GSEA analysis of GBM DEGs (GBM/Control) showed a significant enrichment of many gene sets associated to TGF-β signaling (Supplementary Figure 3)

3) FMOD and TGM2 RNA levels are more elevated in DGCs than in GSCs, and decreased upon treatment of cells with the TGF-β inhibitor (SB431542) (Figure 1G)

4) FMOD (intracellular and secreted, Figure 1H) and pSMAD (Figure 1I) levels are more elevated in DGCs than in GSCs, and decreased upon treatment of cells with the TGF-β inhibitor SB431542

5) TGF-β1-induced luciferase activities of SBE-Luc and FMOD Promoter-Luc reporters are higher in MGG8-DGCs than in MGG8-GSCs, and are inhibited upon SB431542 treatment (Supplementary Figures 5C, D).

6) pSMAD2 occupancy in FMOD promoter is higher in MGG8-DGCs than in MGG8GSCs and is reduced by SB431542treatment (Figure 1J)

All the above data are presented in pages no. 6 and 7, lines 166-188 in the revised manuscript.

5) The authors bounce between cell lines. I am not clear why the authors use MGC lines for the discovery then use other cell types for all the experiments. The rationale for using LN229 and U251 is particularly weak, as these cells have been extensively cultured in serum. I would drop the studies with these lines or relegate them solely to supplemental figures.

In the revised manuscript, we have included experiments in an intracranial orthotopic tumor mouse model using one human GSC cell-line (MGG8; Figures 6 A-D), and a new murine glioma cell line (DBT-Luc; Supplementary Figure 23) in addition to the previously used murine (AGR53; Figures 5A-G) and human glioma cell lines (U251; Figure 6E –I). This data is described in page no. 12-14, lines no 363 to 402 in the revised manuscript.

6) To address the concerns about culture conditions complicating interpretation of comparing GSC and DGC, I would suggest performing more studies with cell sorting and comparison with the same culture conditions. MGC have been cultured for prolonged periods and may resemble cell lines.

We agree in principle with the reviewer. However, we cultured GSCs as neurospheres in neural stem cell media supplemented with growth factors so that the enrichment in CD133-positive cells of the neurospheres is maintained as reported by Suva et al., Cell 157, 580-594, 2014. The enrichment of CD133-positive cells was verified at regular intervals of culturing.

Further, we now show that the MGG8 GSC neurospheres used in our experiments (Figure 6) are indeed enriched in CD133-positive cells. This data is described in page no. 13, lines no 389 to 391 in the revised manuscript.

Moreover, the “DGC-specific” expression of FMOD was demonstrated when both GSCs and DGCs are grown as monolayers in Geltrex-coated cell culture dishes. More detail is given in answer to comment no # 2 of the reviewer

7) The studies only use single knockdown approaches (sometimes siRNA, sometimes shRNA, etc.). It would be helpful to demonstrate better controls and rescue.

In the previous version of the manuscript, siRNA was used to silence FMOD expression in U251 cells while shRNA was used in MGG8-DGCs for the purpose of collecting conditioned media to set up in vitro angiogenesis experiments. In the revised manuscript, silencing of FMOD was achieved by using shRNA in U251 cells and siRNA in MGG8-DGCs. (Supplementary Figures 13 B and E). This data is described in page no. 8, lines no 235 to 238, and page no. 9, lines no. 248-251 in the revised manuscript.

We had already illustrated the “rescue” experiments in U251/siFMOD and MGG8DGC/shFMOD cells (see Figures 3C, D) in the first version of the manuscript. In the revised manuscript, additional rescue experiments were carried out in human brain-derived primary endothelial cells (HBMECs) and mouse brain-derived immortalized endothelial cells (bEnd.3) (see Figure 3 E and F; Supplementary Figure 14F). In all these cellular modles, the reduced angiogenic network formation in presence of CM from FMOD-silenced cells is rescued by the exogenous addition of rhFMOD. This data is described in pages no. 8 and 9, lines no 224 to 257 in the revised manuscript.

8) I would strongly suggest consideration of FMOD analysis in the absence of culture. They should consider double staining of tumor specimens for stem/differentiation markers and FMOD. Also, FMOD expression should be near blood vessels and pFAK and Hes1.

To address this question, we used tissue samples from various intracranial tumor models (human and murine) to show that FMOD is indeed colocalized with GFAP (used as differentiation marker), to validate our initial finding that FMOD is “solely” secreted by the DGCs. These new data are illustrated on Supplementary Figures 24-26, described in page no. 14, lines no. 403 to 414 in the revised manuscript.

We also show a co-localization of CD31 and FMOD, CD31 and pFAK, CD31 and JAG1, and CD31 and HES1 in all three tumor models (Supplementary Figures 28A-C). This data is described in pages no. 14 and 15, lines no. 428 to 434 in the revised manuscript.

9) It would be helpful to consider the choice of lines with the studies of FMOD. FMOD appears to be lower in proneural tumors, which is not surprising due to the association with TGF-β. It would be important to consider the subtype of all the lines used.

We agree with the reviewer that FMOD expression is not associated with proneural tumors. We show in the revised manuscript that the expression of FMOD and the enrichment of TGFβ signaling gene set are more pronounced in mesenchymal subtype GBM (Supplementary Figures 5A and B). We also show that DGCs used in this study, which exhibit high FMOD expression, are also of the mesenchymal subtype, with an enrichment of genes of the TGF-β signaling pathway (Supplementary Figure 4). This data is described in pages no. 6 and 7, lines no. 166 to 188 in the revised manuscript.

10) ST1 cells are not ideal as endothelial cells, as they have been immortalized, which may change the signaling.

To address this issue, we have used two endothelial cells lines of brain origin, primary human brain microvascular endothelial cells (HBMECs) and murine immortalized brain endothelial cells (B.End3). Our results show that CM of MGG8-DGC/shNT cells is more efficient to promote angiogenic network formation by bEnd.3 and HBMECs than CM of MGG8DGC/shFMOD cells (Figures 3E and F; Supplementary Figure 14F), reminiscent of our initial observations using ST1 cells. Again, the effect of FMOD silencing was rescued by adding rhFMOD to both cell types (Figures 3 E and F; Supplementary Figure 14F). This indicates that the pro-angiogenic effect of FMOD is conserved across endothelial cells, irrespective of their tissue of origin, and confirms the validity of our initial findings using the human pulmonary microvascular endothelial cells (ST1). See page no. 9, lines no. 253 to 257 for more detail in the revised manuscript.

11) It would seem that the upstream regulation of FMOD should be better investigated to understand why DGC, not GSC, express FMODAs suggested by the reviewer, we carried out several experiments to confirm that FMOD expression in DGCs is under the control of TGFβ signalling. The detail is provided in the answer to comment #3 above.12) The key experiments would seem to be determining if over expression of FMOD in GSC could replace DGC with in vivo growth. This is the more important finding. I was particularly concerned with the design of Figure 6. The authors assume that the effects of FMOD are due to the DGC, but there is no direct evidence to this effect. I was not even sure why there was not greater study of the xenografts for expression of FMOD and the other signaling molecules.

While we show that FMOD is an important mediator of angiogenesis induced by DGCs, thereby contributing to tumor growth, DGCs may also influence tumor growth through FMODindependent mechanisms. For example, Wang et al. showed a reciprocal signaling between GSCs and DGCs in promoting tumor growth whereby DGC-secreted BDNF acts on NTRK2 receptors on the GSC membrane to activate self-renewal and activation of GSCs. (Wang et al., Cell Stem Cell, 22, 514-528, 2018). Hence, it is unlikely that overexpressing FMOD in GSCs would replace DGCs in tumor growth in vivo.

To address the reviewer’s other concerns, we utilized tissue samples from our various intracranial tumor models (human and murine) to show that FMOD is colocalized with GFAP (used as a differentiation marker) and thus validating our initial finding that FMOD is “solely” secreted by the DGCs in vitro. These data are illustrated on Supplementary Figures 24-26. We further demonstrate that the small tumors formed in FMOD-silenced conditions show GFAP staining, suggesting that differentiation has indeed occurred in the absence of FMOD, which supports our in vitro data indicating that FMOD is not needed for GSC differentiation to DGC.

Further analysis showed reduced angiogenesis in the small tumors formed in FMOD-silenced conditions (Figures 7A-L; Supplementary Figures 27 A-I). From these results, we conclude that compromised angiogenesis is indeed the main reason why tumor growth is reduced in FMOD-silenced condition. We also show the co-localization of CD31 and FMOD, CD31 and pFAK, CD31 and JAG1, and CD31 and HES1 in all three tumor models used in our study, which confirms (in addition to in vitro data) the activation of integrin-Notch signaling in FMOD-treated endothelial cells (Supplementary Figures 28 A-C). See pages no. 14 and 15, lines no. 403 to 434 for more detail in the revised manuscript

13) FMOD appears to be prognostic. This has been reported by the authors but could be better developed.

We have obtained the following data highlighting the prognostic significance of FMOD:

1) Significant upregulation of FMOD, JAG1, and HES1 in GBM in 5 data sets (Supplementary Figure 21 A-C)

2) Correlation between FMOD and HES1 transcripts and between FMOD and JAG1 transcripts in 8 data sets (Supplementary Figures 21 D and E)

3) Association of high FMOD transcript levels with poor prognosis in 5 data sets (Supplementary Figure 22 A)

4) Association of FMOD promoter hypomethylation with poor prognosis in 4 data sets (Supplementary Figure 22 B).

See page no. 12, lines no. 353 to 362 for more detail in the revised manuscript.

14) With FMOD already being reported to be connected to invasion, I was not sure why this was not examined. Also, the effects of FMOD on the immune system should be considered.

To address this issue, we performed invasion assays of ST1 cells treated with BSA vs. rhFMOD or CM of either MGG8-DGC/shNT or MGG8-DGC/shFMOD cells. We show that the presence of FMOD favors invasion of the ST1 cells (Supplementary Figures 14 C and D). See page no. 9, lines no. 251 to 253 for more detail in the revised manuscript.

We agree with the reviewer that investigating the effect of FMOD on the immune system would be of great interest and certainly warrants further exploration. However, we have been successful in addressing almost all comments by producing an important amount of new data both in vitro and in vivo. We would like to address this point in the future. We hope that the reviewer will share our opinion that it is beyond the main scoop of the current article.

15) The summary figure appears premature, as there are a number of molecules that have not been tested and the design of the Notch and KAK studies were not ideal. The authors should have engineered endothelial cells to have activated Notch and FAK to see if this rescues the effects of loss of FMOD in DGC.

To address the reviewer’s concerns, we carried out the following experiments in the revised manuscript.

1) We showed that αVβ1 and α6β1integrins play an important role in rhFMOD-mediated induction of pFAK levels (Supplementary Figures 18 D-I).

2) Pre-treatment of ST1 cells with FAK inhibitor or a Src inhibitor (PP2) reduced significantly the rhFMOD-mediated activation of CSL-Luc and HES-Luc (Supplementary Figures 19 AD), and induction of HES1 transcript and protein in ST1 cells (Supplementary Figure 19 E-

H).

3) Pre-treatment of ST1 cells with FAK inhibitor or PP2 (a Src inhibitor) significantly reduced the activation of JAG1 by rhFMOD (Supplementary Figures 20 B and C).

4) Silencing JAG1 in endothelial cells inhibited the ability of rhFMOD to activate CSL-Luc and HES-Luc (Supplementary Figures 20 D and E), increase HES1 transcript and protein levels, and induce angiogenic network formation by the ST1 cells (Figures 4 I-L).

5) We have identified KLF8 as the FAK-activated transcription factor responsible for JAG1 activation in rhFMOD-treated endothelial cells (Supplementary Figures 20 G and H).

6) We have also engineered ST1 cells exhibiting constitutively active Notch signaling. We stably expressed the Notch-intracellular domain (NICD) or the Vector Control (VC) in ST1 (ST1/NICD, ST1/VC) cells. (Supplementary Figure 17A and B). ST1/NICD cells showed higher capacity to form angiogenic networks by themselves, compared with ST1/VC cells (Supplementary Figures 17C and D). ST1 cells were then be subjected to angiogenesis network formation assay in the presence or absence MGG8-DGC/shNT vs. MGG8DGC/shFMOD CM. We observed that the lack of FMOD in MGG8-DGC/shFMOD CM caused a significant decrease in network formation by ST1/VC cells but not by ST1/NICD cells (Supplementary Figures 17 C and D). These results confirm that Notch activation is an essential step in FMOD-induced angiogenesis. *See pages no. 9-12, lines no. 272 to 362 for more detail in the revised manuscript.*

7) We also show co-localization of CD31 with FMOD, pFAK, JAG1 and HES1 in the three tumor models used in our study, which confirms (in addition to in vitro data) the activation for integrin-Notch signalling in FMOD-treated endothelial cells (Supplementary Figures 28 AC). *See page no. 14, lines no. 428 to 434 for more detail in the revised manuscript.*

Minor points:1) The data presentation is not professional. The figures do not comply to Cell journal standards.

Efforts were made to make appropriate changes in the Figures and to improve their readability.

2) Better statistical testing with correction for multiple comparisons should be performed.

We have now systematically performed ANOVA in experiments where multiple samples were compared.

Answer to comments from Reviewer 31) The authors use 10% FBS to induce GSC differentiation into DGC, and the corresponding two types of conditioned medium were harvested and analyzed by mass spectrometry. It is unclear if these two cell types were cultured in same medium (stem cell medium or normal cancer cell medium?). Furthermore, FMOD was identified to be enriched in DGC-conditioned medium, which could be simply due to the effects of floating (cancer stem cells) vs adherent cells (differentiated cells). As this is the foundation of the whole study, the authors need to verify it in a well-controlled system, e.g., glioma stem cells (CD133+) vs non-stem glioma cells (CD133-), both cultured as neuro spheres, from the same patients. Since many datasets of global gene expression in GSCs and DGCs have been published, the authors could easily verify the specific mRNA expression of FMOD in DGCs.

Overall, we agree with the reviewer. However, we cultured GSCs as neurospheres in neural stem cell media supplemented with growth factors so that the enrichment of CD133-positive cells in neurospheres in was maintained (see Suva et al., Cell 157, 580-594, 2014). We also verify this point at regular intervals of culturing and show that the MGG8-GSC neurospheres used in our experiments (Figure 6A) are indeed enriched in CD133-positive cells.

Moreover, the DGC-specific expression of FMOD is also shown when both GSCs and DGCs are grown as monolayers in Geltrex-coated cell culture dishes. More detail is given in the answer to comment #2 of Reviewer #2.

2) Figure 3. The in vitro angiogenesis analysis is weak, solely depending on tube formation assay. The results need to be further verified by EC proliferation and migration assays.

To address the reviewer’s comment, we have included a number of new data:

1) We show that FMOD (rhFMOD or FMOD coming from MGG8-DGC/shNT CM) induces migration, invasion, but not proliferation of ST1 cells (Supplementary Figures 14A-E). See page no. 9, lines no. 249 to 251 for more detail in the revised manuscript.

2) Using two additional endothelial cell lines of brain origin (namely primary human brain microvascular endothelial cells (HBMECs) and murine immortalized brain endothelial cells (b.End3), we show that CM from MGG8-DGC/shNT cells promotes formation of angiogenic networks by HBMECs (Figures 3E and F) and B.End.3 (Supplementary Figure 14F) more efficiently than CM from MGG8-DGC/shFMOD cells. The effect of FMOD silencing was rescued by adding rhFMOD to both cell types (Figures 3 E and F; Supplementary Figure 14F). This indicates that the pro-angiogenic effect of FMOD is conserved across endothelial cells, irrespective of their tissue of origin, and confirms the validity of our initial findings using human pulmonary microvascular endothelial cells (ST1). See page no. 9, lines no. 253 to 258 for more detail in the revised manuscript.

3) The small size tumors formed by AGR53-GSC/miRFMOD cells after doxycycline treatment showed reduced staining for CD31 and vWF (von Willebrand factor) compared to tumors formed by AGR53-GSC/miRNT cells (Figures 7A-C; Supplementary Figures 27 A-C). Likewise, tumors formed by DBT-Luc GSC/miRFMOD cells in doxycycline-treated mice showed significantly reduced CD31 staining, compared with those developed by DBT-Luc GSC/miRFMOD cells in the absence of doxycycline treatment (Supplementary Figures 27 D-F). Like murine glioma tumors, tumors developed by MGG8-GSC/shFMOD cells showed reduced CD31 staining compared to tumors formed by MGG8-GSC/shNT cells (Figures 7 DF). We also tested the extent of blood vessel formation by TDECs (Tumor-derived endothelial cells). In all the three tumor models used, blood vessels formed by TDECs were significantly reduced in tumors formed in FMOD-silenced condition (Figures 7G-L; Supplementary Figures 27 G- I). See page no. 14, lines no. 414 to 428 for more detail in the revised manuscript.

These findings confirm our results in vitro, where the absence of FMOD decreased the ability of host-derived endothelial cells and TDECs to form angiogenic networks (Figure 3; Supplementary Figures 13-15). From these results, we conclude that angiogenesis induced by DGCsecreted FMOD is essential for glioma tumor growth.

4) We also show the co-localization of CD31 and FMOD, pFAK, JAG1, and HES1 in all the three tumor models used, which confirms (in addition to in vitro data) the activation of integrin-Notch signaling in FMOD-treated endothelial cells (Supplementary Figures 28 A-C). See page no. 14, lines no. 428 to 434 for more detail in the revised manuscript.

3) The mechanism study is relatively weak. The authors should identify the integrin that interacts with FMOD and test its role in FMOD-induced angiogenesis. To support the proposed model (Figure 7), the authors may define the precise mechanism by which FMOD induces FAK/Src activation and Jag1 expression, such as assays for dependency of Src/FAK on Jag1 expression, transcription factor/promoter binding, and transcriptional activation.

To address the reviewer’s comment, the following data have been included in the revised manuscript.

1) With respect to integrin signaling, we show that ITGA6, ITGAV and ITGB1 are highly expressed in endothelial cells (See supplementary information); Silencing experiments proved that all three integrins are important for the increase in pFAK levels induced by rhFMOD. This suggests the involvement of αVβ1 and α6β1 integrins in rhFMOD effects in endothelial cells (Supplementary Figures 18 D-I).

2) Pre-treatment of ST1 cells with FAK inhibitor or PP2 (Src inhibitor) significantly reduced the rhFMOD-mediated activation of CSL-Luc and HES-Luc (Supplementary Figures 19 AD), as well as induction of HES1 transcript and protein in ST1 cells (Supplementary Figures 19 E-H).

3) Pre-treatment of ST1 cells with FAK inhibitor or PP2 reduced the activation of JAG1 by rhFMOD (Supplementary Figures 20 B and C).

4) Silencing JAG1 in endothelial cells inhibited the ability of rhFMOD to activate CSL-Luc and HES-Luc (Supplementary Figures 20 D and E), induce HES1 transcript and protein, and induce angiogenic network formation by the ST1 cells (Figures 4 I-L).

5) We identified KLF8 as the FAK-activated transcription factor responsible for JAG1 activation in rhFMOD-treated endothelial cells (Supplementary Figures 20 G and H).

6) We engineered ST1 cells exhibiting constitutively active Notch signaling. We stably expressed the Notch-intracellular domain (NICD) or the Vector Control (VC) in ST1 (ST1/NICD, ST1/VC) cells (Supplementary Figures 17A and B). ST1/NICD cells showed higher capacity to form angiogenic networks by themselves, compared with ST1/VC cells (Supplementary Figures 17C and D). ST1 cells were then be subjected to in vitro angiogenesis network formation assay in the presence or absence of MGG8-DGC/shNT or MGG8-DGC/shFMOD CM. We observed that the lack of FMOD in MGG8-DGC/shFMOD CM caused a significant decrease in network formation by ST1/VC cells but not by ST1/NICD cells (Supplementary Figures 17 C and D). These results confirm that Notch activation is an essential step in FMODinduced angiogenesis. See pages no. 9-12, lines no 272 to 362 for more detail in the revised manuscript.

4) The authors should validate their proposed molecular mechanism in the in vivo study, such as FMOD-induced and endothelial Jak1 expression in tumor ECs.

We show a co-localization of CD31 and FMOD, pFAK, JAG1, and HES1 in all three tumor models used in the studies, which confirms (in addition to in vitro data) the activation for integrin-Notch signaling in FMOD-treated endothelial cells (Supplementary Figures 28 A-C).

See page no. 14, lines no. 428 to 434 for more detail in the revised manuscript.

Other concerns:1. Numerous mistakes/typos in references, e.g., Wang et al., 2918 (from future?) and Mondal et al., 2017 (could be seen in the ref list) in Page 6 and etc.

We carefully proofread the entire manuscript (including references) and corrected the mistakes.

2. Figure S1. The shRNA knockdown efficiency needs to be shown.

The knockdown of FMOD in MGG8 GSCs and DGCs is shown in the revised manuscript (Supplementary Figure 6A). In addition, FMOD-silencing in tumor models is shown on Figures 5 and 6, and Supplementary Figures 23-26.

3) Figure S2.D.E. The results of FMOD knockdown were not consistent in S2D and S2E, or the lanes were wrongly labeled.

This mistake has been rectified and shown in Supplementary Figure 7 of the revised manuscript.

4) Figure S5. Considering several patient-derived GSCs are available in the authors' laboratory, why authors use U251 and mouse GSCs to test the role of FMOD in stemness and differentiation?

In the revised manuscript, we have included experiments in an intracranial orthotopic tumor model in the mouse using one human GSC cell-line (MGG8; Figures 6 A-D), and an additional murine glioma cell line (DBT-Luc; Supplementary Figure 23) in addition to existing experiments using murine (AGR53; Figures 5A-G) and human (U251; Figures 6E–I) glioma cell lines. This data is described in pages no. 12-14, lines no. 363 to 402.

5) Figure S10. The in vivo angiogenesis data is important for the whole study, and, therefore, is suggested to move to main figure. Furthermore, the main findings need to validate by additional assays, such as vessel perfusion or tumor hypoxia. In addition, the changes in representative images, but not in the quantified data, are not impressive. Additional staining with CD31 immunofluorescence would strengthen the results.

To address this concern, we have carried out the following experiments in the revised manuscript.

1) We have done additional angiogenesis-related experiments in vivo and the results are illustrated on Figure 7 and Supplementary Figure 27.

2) In the revised manuscript, we also demonstrate that FMOD induces angiogenic network formation by tumor-derived endothelial cells (TDECs) both in vitro and in vivo.

in vitro, we demonstrate that MGG8-DGCs and U87 cells readily differentiate to form endothelial cells (tumor-derived endothelial cells; TDECs) when grown in M199 media, which correlated with increased CD31 expression and decreased FMOD expression. In both cases, TDECs were induced to form angiogenic networks by the presence of rhFMOD. These new data are presented on Figures 3G-J and Supplementary Figure 15.

in vivo, we demonstrate that large tumors formed by MGG8-GSC, AGR53-GSC, and DBTLuc-GSC in FMOD+ condition show the presence of TDECs, as evidenced by significant colocalization of CD31 and GFAP (for MGG8), GFP (for AGR53 and DBT-Luc) on the blood vessels. This was significantly reduced in small tumors formed by the above cells in FMODsilenced conditions (Figures 7 G-L; Supplementary Figures 27 G-I).